# Transcription arrest induces formation of RNA granules in mitochondria

Katja G Hansen , Autum Baxter-Koenigs, Caroline AM Weiss, Erik McShane , L Stirling Churchman

**Mitochondrial gene expression regulation is required for the biogenesis of oxidative phosphorylation (OXPHOS) complexes, yet the spatial organization of mitochondrial RNAs (mt-RNAs) remains unknown. Here, we investigated the spatial distribution of mt-RNAs during various cellular stresses using single-molecule RNA-FISH. We discovered that transcription inhibition leads to the formation of distinct RNA granules within mitochondria, which we term inhibition granules. These structures differ from canonical mitochondrial RNA granules and form in response to multiple transcription arrest conditions, including ethidium bromide treatment, specific inhibition or stalling of the mitochondrial RNA polymerase, and depletion of the SUV3 helicase. Inhibition granules appear to stabilize certain mt-mRNAs during prolonged transcription inhibition. This phenomenon coincides with an imbalance in OXPHOS complex expression, where mitochondrial-encoded transcripts decrease while nuclear-encoded subunits remain stable. We found that cells recover from transcription inhibition via resolving the granules, restarting transcription, and repopulating the mitochondrial network with mt-mRNAs within hours. We suggest that inhibition granules may act as a reservoir to help overcome OXPHOS imbalance during recovery from transcription arrest.**

## Introduction

Mitochondria fulfill numerous essential functions to maintain cellular homeostasis. One of the key functions is ATP production through oxidative phosphorylation (OXPHOS) via the OXPHOS complexes. Dysfunction of OXPHOS is associated with diseases, such as encephalopathies, Leigh syndrome, cancer (Thompson Legault et al, 2015; Reznik et al, 2017; Ghezzi & Zeviani, 2018), and plays a key role in cellular aging (Houtkooper et al, 2013; Lima et al, 2022).

Mitochondria contain their own genome that is present in between 100 and 1,000s of copies per cell. The mitochondrial DNA (mtDNA) encodes for 13 OXPHOS subunits, 22 tRNAs, and 2 rRNAs; the residual mitochondrial proteome, including the remaining OXPHOS subunits, is nuclear-encoded and needs to be imported. In order to build functional OXPHOS complexes, gene expression has to be coordinated and balanced across compartments (Gustafsson et al, 2016; Isaac et al, 2018; Soto et al, 2022).

Mitochondrial gene expression is regulated by numerous factors, allowing for the differential control of individual mtDNA molecules within the mitochondrial network. Despite the high copy number of mtDNA in cells, most mitochondrial nucleoids are tightly compacted (Brüser et al, 2021; Isaac et al, 2024), inactive for transcription (Brüser et al, 2021) or replication (Lewis et al, 2016). The transcriptional state varies across the mitochondrial network (Brüser et al, 2021). In addition, the mitochondrial RNA polymerase POLRMT can be reversibly inhibited by binding to the noncoding 7S RNA. This mechanism is thought to enable fine-tuned temporal and spatial control of POLRMT activity across different mtDNA molecules in the network (Zhu et al, 2022). Together, these regulatory mechanisms allow for dynamic and localized control of mitochondrial gene expression.

Similar to the nucleolus, mitochondrial RNA granules (MRGs) serve as specialized sites for RNA processing and ribosome assembly within mitochondria (Antonicka & Shoubridge, 2015; Jourdain et al, 2015; Popow et al, 2015; Tu & Barrientos, 2015; Boehm et al, 2017; Rey et al, 2020; Ohkubo et al, 2021; Xavier & Martinou, 2021). MRGs contain unprocessed polycistronic RNAs and key RNA-binding proteins (RBPs) such as GRSF1 and FASTKD2. Within MRGs, nascent RNA is processed into mRNAs, tRNAs, and rRNAs (Antonicka et al, 2013; Jourdain et al, 2013, 2015, 2016; Antonicka & Shoubridge, 2015; Popow et al, 2015; Rey et al, 2020). In addition to MRGs, other distinct RNA-containing structures have been identified within mitochondria. These include D-foci, which are sites of RNA degradation and contain the mitochondrial degradosome, a dimer formed by the exoribonuclease PNPT1 (PNPase) and the DNA helicase SUV3 (Borowski et al, 2013; Xavier & Martinou, 2021). Double-stranded RNA (dsRNA) in mitochondria is also found in distinct foci (Dhir et al, 2018). Mitochondria may also form stress-induced RNA granules, similar to the well-characterized cytoplasmic stress granules that sequester translation initiation

---

Department of Genetics, Blavatnik Institute, Harvard Medical School, Boston, MA, USA

Correspondence: churchman@genetics.med.harvard.edu
Katja G Hansen's present address is Department of Cell Biology, University of Kaiserslautern-Landau, Kaiserslautern, Germany
Caroline AM Weiss's present address is Department of Proteomics and Signal Transduction, Max Planck Institute of Biochemistry, Martinsried, Germany

complexes and mRNAs during cellular stress (Anderson & Kedersha, 2009). Recent studies show the formation of mitochondrial stress granules or stress granule–like structures during certain conditions (Sun et al 2023; Begeman et al, 2025). However, the full range of stress conditions that induce these mitochondrial structures, their RNA composition, and their functions remain to be fully elucidated.

In this study, we investigated the formation of stress-induced RNA granules inside mitochondria. Using single-molecule RNA fluorescence in situ hybridization (RNA-FISH), we examined the spatial distribution of mt-RNAs under various stress conditions. Our findings reveal that transcription inhibition triggers the formation of novel RNA granules, which we have termed "inhibition granules." These structures are distinct from the canonical mitochondrial RNA processing granules (MRGs). We demonstrate that all mt-mRNAs localize to these inhibition granules, and importantly, these structures appear to stabilize a subset of the transcripts. Despite this protective effect, overall mt-RNA abundance decreases during transcription inhibition, leading to an imbalance in the expression of OXPHOS components. Interestingly, cells can recover from transcription inhibition by reinitiating transcription and redistributing mt-mRNAs throughout the mitochondrial network within hours upon stress release. Based on these observations, we hypothesize that inhibition granules serve as temporary RNA reservoirs, which may help to restore OXPHOS balance and mitochondrial function upon stress alleviation.

## Results

### Ethidium bromide treatment leads to the formation of RNA granules

In the cytosol, stress granules (SG) form in response to stress, a process that is poorly understood in mitochondria. Therefore, we aimed to investigate the spatial distribution of mt-mRNAs during steady-state and stress conditions using single-molecule RNA-FISH. However, some mt-mRNAs are only a few hundred base pairs long (e.g., MT-ND3 is 346 nucleotides long), rendering their visualization difficult. Hence, we employed SABER-FISH (signal amplification by nucleotide exchange reaction) (Kishi et al, 2019) to detect mt-mRNAs. SABER-FISH uses nucleotide exchange reactions to elongate primer sequences in an oligo pool creating adapters of several hundred base pairs in length. This drastically increases the detectable signals as a high number of small fluorescent oligos are associated with the adaptors (Kishi et al, 2019). We robustly detect mitochondrial-encoded RNAs in U2-OS cells using SABER-FISH (Figs 1A and B and S1A). Our data shows that mt-mRNAs are highly abundant and distributed throughout the mitochondrial network.

Upon successful implementation of the SABER-FISH technology on all mt-mRNAs, we aimed to investigate possible changes in their spatial distribution during cellular stress. To cover a broad range of various stress inducers, we analyzed mt-mRNA localization after treatment with: the proteasomal inhibitor bortezomib known to cause stress granule formation in the cytosol (Fournier et al, 2010), the complex III inhibitor antimycin A, ethidium bromide, which intercalates into mtDNA and inhibits DNA replication and

transcription, and chloramphenicol, a mitochondrial translation inhibitor (Figs 1C and S1B and C). To find ethidium bromide concentrations that would lead to a strong transcription inhibition, we used qRT-PCR to test the decrease in RNA abundance after 5 h of treatment in HeLa cells (Fig S1D). We treated U2-OS cells for 5 h (Figs 1C and S1C) or 24 h (Fig S1B) and probed for either with MT-CO1 or in addition with MT-ND2. Of these treatments, only EtBr led to a profound visible change in RNA localization and seemingly co-accumulation of RNAs compared with its control, where the labeled transcripts formed clusters resembling RNA granules (Fig 1D). The formation of RNA clusters was robust across various concentrations of EtBr (Fig S2A) and occurred also in HeLa cells (Fig S2B).

### RNA granules form under multiple mitochondrial transcriptional perturbations

To further characterize the EtBr-induced granules, we conducted a comprehensive analysis of all mitochondrial-encoded OXPHOS subunits. Using RNA-FISH probes specific to each mitochondrial transcript and to the antisense RNA of MT-CO1, we observed that all mitochondrial mRNAs exhibited a granular distribution pattern similar to that of MT-CO1 and MT-ND2. This consistent behavior across all mitochondrial transcripts suggests a global response of mitochondrial RNAs to transcription inhibition, rather than a transcript-specific phenomenon (Figs 2A and S3A–C). To investigate the timing of granule formation, we performed a time course experiment, which showed that the first granules were visible after 2 h of EtBr treatment (Fig 2B). To quantify RNA granule formation after EtBr treatment, we measured the fraction of cellular area occupied by RNA, calculated as the segmented area of the RNA signal as a fraction of the total cell area, a measure used in the analysis of granule formation in the cytosol (Mateju et al, 2020) (Fig 2C). The fraction of cellular area occupied in the EtBr-treated samples decreased for most of the transcripts, indicating that RNAs are less distributed and form more granules, in both biological replicates (Figs 2D and S4A). As a second measure, we analyzed the signal distribution as a histogram within cells. Granule formation, leading to more stable transcripts, should lead to a tail right of the center of the mass in a histogram, seen as numbers greater than 0. The distribution shifted for most of the transcripts to the right, confirming the result of the fraction of cellular area occupied measurement (Figs 2E and S4C). To confirm that granule formation is directly triggered by transcription arrest, rather than by potential off-target effects of ethidium bromide (EtBr) (King & Attardi, 1989; Toompuu et al, 2018), we investigated two additional conditions that inhibit mitochondrial transcription through distinct mechanisms. First, we treated cells with IMT1B, a specific inhibitor of POLRMT (Bonekamp et al, 2020). Investigating the spatial distribution of two representative mt-mRNAs, MT-ND5 and MT-CO3, revealed granule formation similar to that observed with EtBr treatment (Figs 2F–I and S4B and D). Second, we used the adenosine analog 2′-C-methyladenosine (2′-CMA), which leads to stalling of POLRMT and a strong decrease in transcription (Fig S5A). Again, we observed visual granule formation of MT-CO3 and MT-ND5.

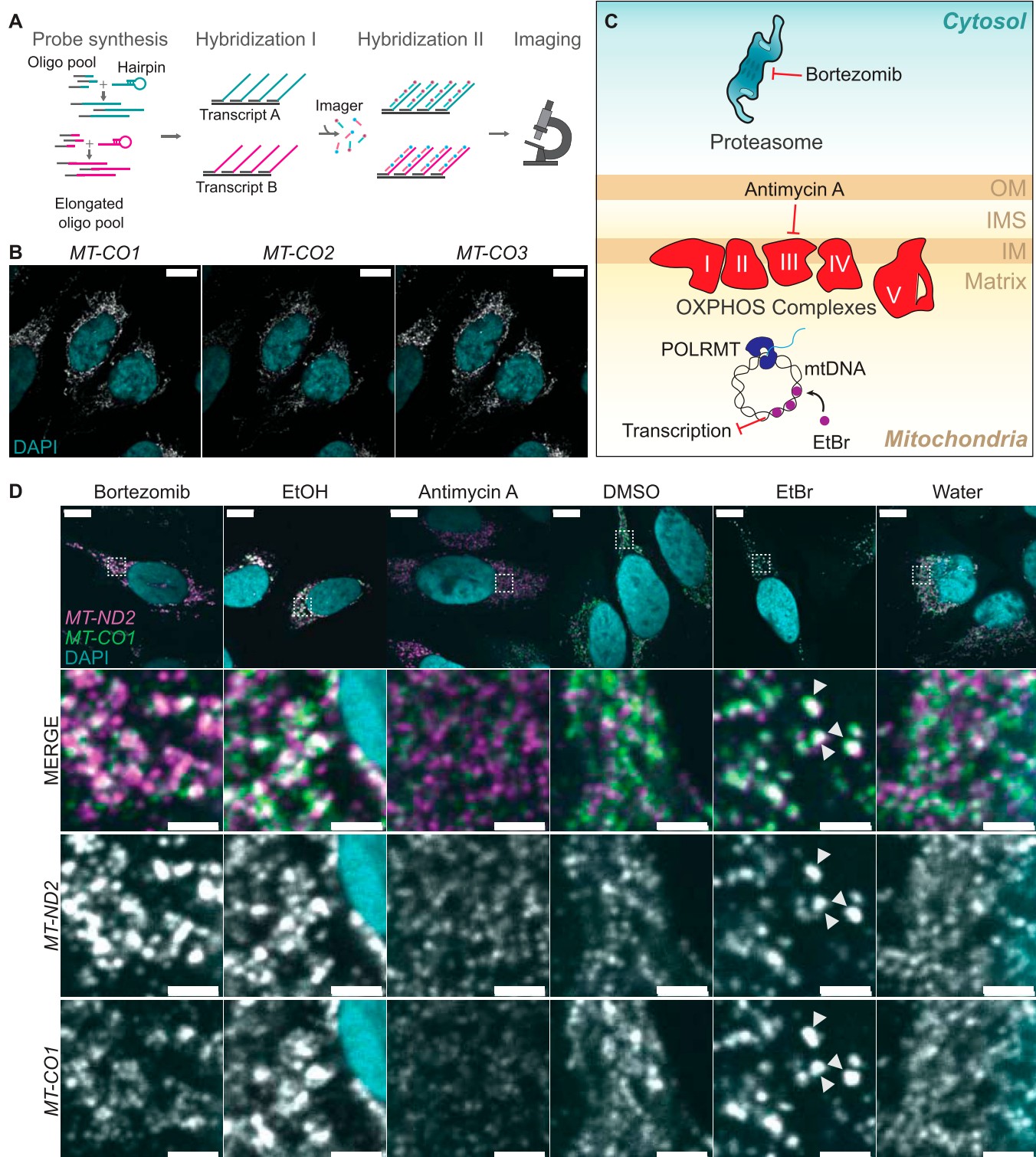

**Figure 1.  Ethidium bromide treatment leads to changes in the spatial distribution of the mitochondrial transcripts *MT-CO1* and *MT-ND2*.**
**(A)** Workflow of SABER-FISH. Oligo pools are elongated with the help of hairpins creating long overhangs. Pink and turquoise colors represent different sequences. In the first hybridization, probes bind to transcripts of interests. In the second hybridization, imager probes with fluorophores bind to the complementary overhangs followed by imaging. **(B)** Confocal fluorescence microscopy of multiplexed SABER-FISH of three different transcripts in U2-OS cells. The complex IV subunit transcripts *MT-CO1*, *MT-CO2*, and *MT-CO3* were labeled. The scale bars represent 10 μm. **(C)** Scheme showing acting sites of the used drugs. Bortezomib inhibits the proteasome, antimycin A blocks oxidative phosphorylation complex III, and ethidium bromide (EtBr) intercalates into the mitochondrial DNA and subsequently inhibits transcription. **(D)** SABER-FISH for *MT-CO1* (green in the MERGE panel) and *MT-ND2* (magenta in the MERGE panel) after different drug treatments. Cells were treated with either 100 μM

Collectively, these results demonstrate that inhibition granules form in response to transcription arrest, whether induced pharmacologically or genetically. Thus, the formation of these granules appears to be a consistent cellular response to decreased mitochondrial transcription.

### Inhibition granules are distinct from RNA processing MRGs

We next asked whether the granules described here are distinct from the canonical MRGs. Two RNA-binding proteins, GRSF1 and FASTKD2, are key for the formation of MRGs (Antonicka et al, 2013; Jourdain et al, 2013; Antonicka & Shoubridge, 2015; Popow et al, 2015; Rey et al, 2020; Xavier & Martinou, 2021). We used immunofluorescence to examine the localization of GRSF1 and FASTKD2, after treatment with EtBr or IMT1B (Fig 3A–D). Under normal conditions, both proteins displayed characteristic foci within the mitochondrial network, consistent with their known localization in MRGs. However, upon transcription inhibition, neither GRSF1 nor FASTKD2 exhibited the clustering observed for mt-mRNAs. Instead, GRSF1 lost its granular distribution, a finding consistent with previous reports that GRSF1 requires active transcription to form MRGs (Antonicka et al, 2013; Jourdain et al, 2013). After IMT1B treatment, FASTKD2 also lost its granular distribution. EtBr treatment showed the same effect, but not as severe as some FASTKD2 remains in potential granules. Recently, it has been found that down-regulation of SUV3 leads to reduced transcription (Zhu et al, 2022). Indeed, when we knockdown SUV3 we see that GRSF1 shows less localization to MRGs (Fig S5B and C). In these conditions, we also observe similar RNA granules as seen after EtBr or IMT1B treatment (Fig S5D). These observations suggest that the RNA-containing structures formed during transcription inhibition are distinct from canonical MRGs that we therefore designate as "inhibition granules."

RNA granule formation is influenced by molecular crowding (André & Evan, 2020; Hutten & Dormann, 2020), which has been reported to increase in the mitochondrial network upon stress as cristae are disturbed (Bulthuis et al, 2023). Hence, a disturbed mitochondrial network could lead to RNA clusters similar to the phenotype we observe. To investigate this possibility, we used mitochondrial markers, the leucine-rich pentatricopeptide repeat–containing protein LRPPRC or TOM20, to determine the integrity of the mitochondrial network after EtBr or IMT1B treatment (Figs 3 and S6A and B). In addition, transmission electron microscopy (TEM) was applied to visualize the crista structure after drug treatment (Fig S6C and D). For EtBr, some cells showed displacements of cristae and swollen mitochondria (Fig S6A–D). However, this was not the case for the more selective IMT1B; here, the mitochondrial network was unchanged after 5 h of treatment

(Fig 3C and D) or even after 24 h of treatment (Fig S6C and D). Thus, the granule formation is not caused by a disturbed mitochondrial network for IMT1B, and could play only a minor role in EtBr.

### Levels of mt-RNAs decrease without changes to nuclear-encoded RNAs

After transcription inhibition, mt-RNAs are expected to decrease with half-lives measured between 20 and 160 min (Gelfand & Attardi, 1981; Nagao et al, 2008; McShane et al, 2024). By quantifying the overall RNA intensities per cell, we observed a strong decrease for most of the tested transcripts in EtBr (Fig S4E)- or IMT1B-treated cells (Fig S4F), reflecting a decrease in RNA abundance.

We next asked whether nuclear gene expression changes in response to lowered mt-mRNA levels. RNA-seq analysis after 5 and 24 h of EtBr treatment of HeLa cells showed that mitochondrial-encoded transcripts decrease severely in abundance (Fig 4A), in line with the microscopy data (Fig S4E). In contrast, levels of mRNAs of nuclear-encoded OXPHOS subunits did not change (Fig 4A), and neither did other gene groups. Thus, transcription arrest leads to an imbalance in OXPHOS expression, which has been shown to be detrimental to cellular homeostasis (Kruse et al, 2008; Kühl et al, 2017; Soto et al, 2022).

### Cells resolve inhibition granules upon stress release

We next investigated whether cells can recover and resolve inhibition granules. We treated cells for 5 h with EtBr and studied RNA distribution during recovery for 4, 5, 6, and 8 h. Interestingly, cells were able to recover as granules disappear after 6 h of stress release, although they began to dissipate hours earlier (Figs 4B and S7A).

We next sought to uncover how cells recover from transcription inhibition. We first measured how total mt-mRNA levels change during recovery after IMT1B treatment. RNA levels increased after 5 h of recovery almost to the same amount of untreated cells (Fig 4C), suggesting that transcription restart contributes to the RNA repopulation of the mitochondrial network. To determine the timing of transcription restart, we performed transient transcription analysis during recovery (Schwalb et al, 2016). We labeled new RNA with 4sU for 10 min of each time point (0–5 h) and analyzed the fraction labeled in a metagene plot. Already after 1 h of recovery, there was a slight increase in the newly made fraction of RNAs measured. The amount of newly made transcripts in 10 min increased further overtime, showing that although transcription restarts early, higher rates of transcription emerge slowly (Figs 4D and S7B).

---

bortezomib, antimycin A, or 2 μg/ml EtBr for 5 h. The respective vehicle control was used and is shown next to the corresponding drug. Images of representative cells are shown. White boxes indicate areas that are shown enlarged underneath. Scale bars of full cell images represent 10 μm and of zoom-in represent 2 μm. White arrows mark clustered RNA in EtBr-treated cells. For all images shown in this study, if not otherwise mentioned, single z-planes were chosen. DAPI staining is shown in cyan and RNA staining in gray. All images in this figure were adjusted to represent the spatial distribution and not differences in intensities.

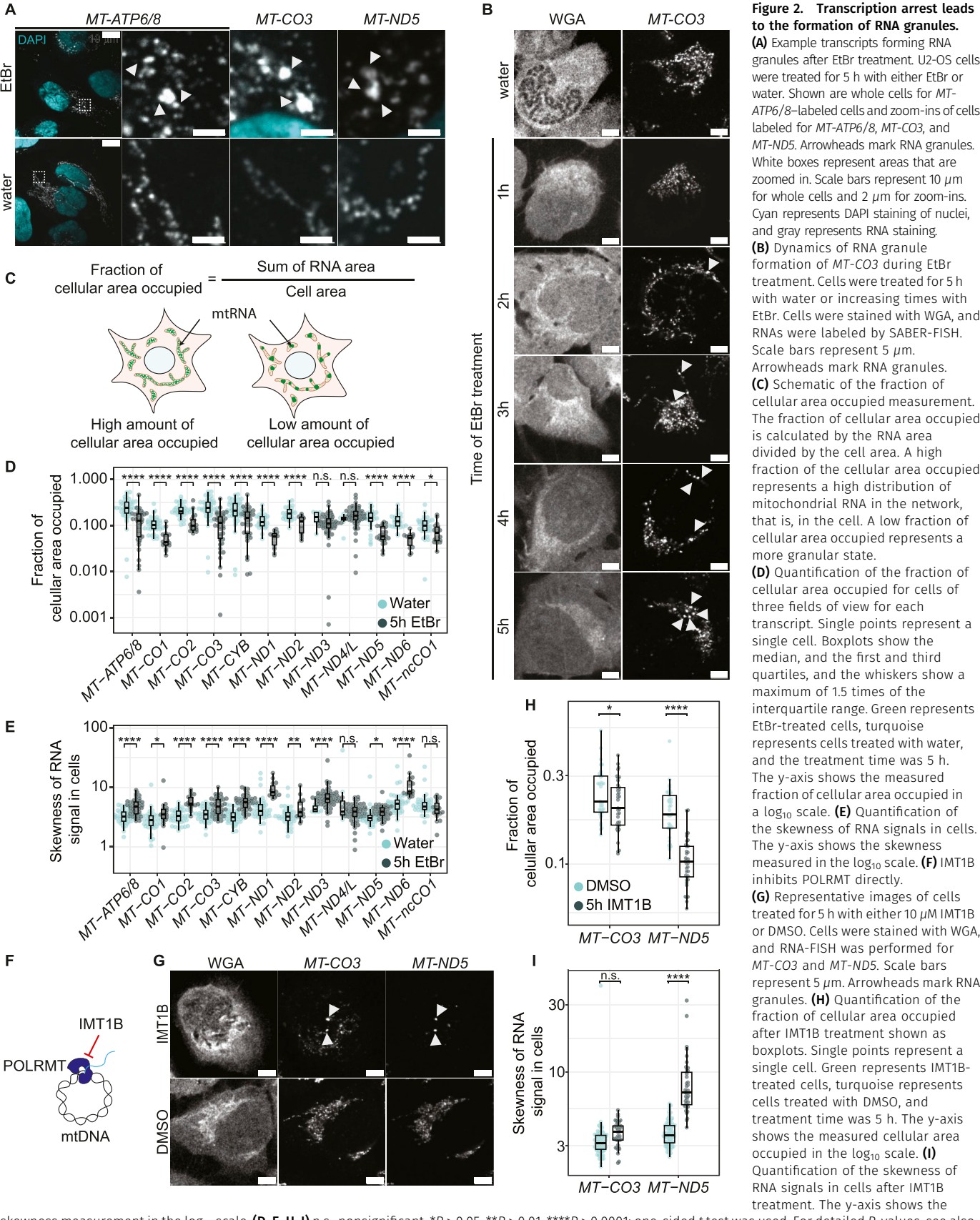

**Figure 2. Transcription arrest leads to the formation of RNA granules.**
**(A)** Example transcripts forming RNA granules after EtBr treatment. U2-OS cells were treated for 5 h with either EtBr or water. Shown are whole cells for *MT-ATP6/8*–labeled cells and zoom-ins of cells labeled for *MT-ATP6/8*, *MT-CO3*, and *MT-ND5*. Arrowheads mark RNA granules. White boxes represent areas that are zoomed in. Scale bars represent 10 μm for whole cells and 2 μm for zoom-ins. Cyan represents DAPI staining of nuclei, and gray represents RNA staining.
**(B)** Dynamics of RNA granule formation of *MT-CO3* during EtBr treatment. Cells were treated for 5 h with water or increasing times with EtBr. Cells were stained with WGA, and RNAs were labeled by SABER-FISH. Scale bars represent 5 μm. Arrowheads mark RNA granules.
**(C)** Schematic of the fraction of cellular area occupied measurement. The fraction of cellular area occupied is calculated by the RNA area divided by the cell area. A high fraction of the cellular area occupied represents a high distribution of mitochondrial RNA in the network, that is, in the cell. A low fraction of cellular area occupied represents a more granular state.
**(D)** Quantification of the fraction of cellular area occupied for cells of three fields of view for each transcript. Single points represent a single cell. Boxplots show the median, and the first and third quartiles, and the whiskers show a maximum of 1.5 times of the interquartile range. Green represents EtBr-treated cells, turquoise represents cells treated with water, and the treatment time was 5 h. The y-axis shows the measured fraction of cellular area occupied in a $\log_{10}$ scale. **(E)** Quantification of the skewness of RNA signals in cells. The y-axis shows the skewness measured in the $\log_{10}$ scale. **(F)** IMT1B inhibits POLRMT directly.
**(G)** Representative images of cells treated for 5 h with either 10 μM IMT1B or DMSO. Cells were stained with WGA, and RNA-FISH was performed for *MT-CO3* and *MT-ND5*. Scale bars represent 5 μm. Arrowheads mark RNA granules. **(H)** Quantification of the fraction of cellular area occupied after IMT1B treatment shown as boxplots. Single points represent a single cell. Green represents IMT1B-treated cells, turquoise represents cells treated with DMSO, and treatment time was 5 h. The y-axis shows the measured cellular area occupied in the $\log_{10}$ scale. **(I)** Quantification of the skewness of RNA signals in cells after IMT1B treatment. The y-axis shows the

skewness measurement in the $\log_{10}$ scale. **(D, E, H, I)** n.s., nonsignificant, *$P > 0.05$, **$P > 0.01$, ****$P > 0.0001$; one-sided *t* test was used. For detailed *P*-values, see also Table S4. All images in this figure were adjusted to represent the spatial distribution and not differences in intensities.

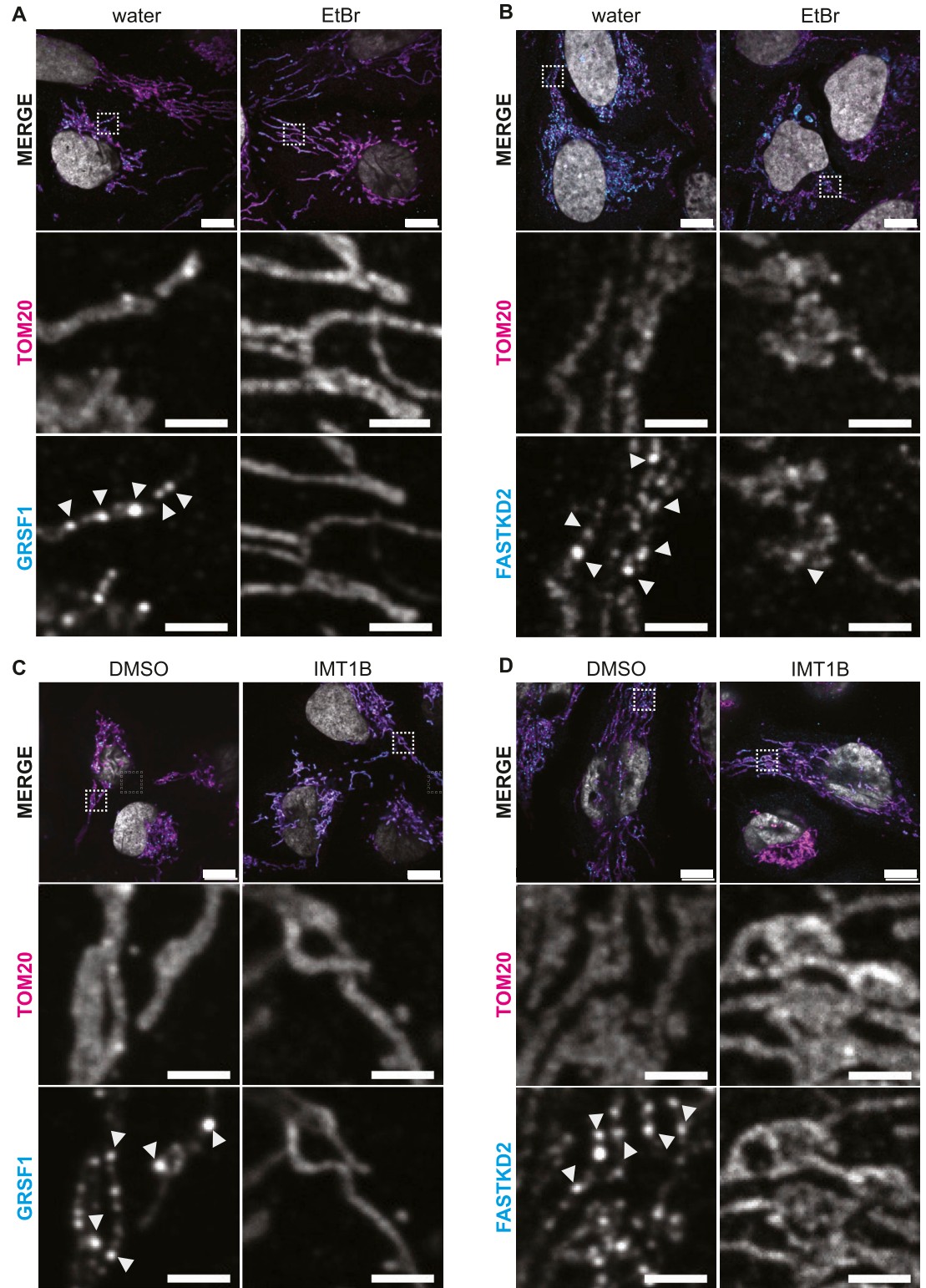

**Figure 3.   Mitochondrial RNA granule markers GRSF1 and FASTKD2 are not part of inhibition granules.**
**(A, C)** Immunofluorescence of cells treated for 5 h with EtBr, water, IMT1B, or DMSO. TOM20 was used as a mitochondrial marker in all panels. TOM20 is shown in magenta in the MERGE panel and GRSF1 in cyan. **(B, D)** Immunofluorescence of cells treated for 5 h with EtBr, water, IMT1B, or DMSO. TOM20 was used as a mitochondrial marker in all panels. TOM20 is shown in magenta in the MERGE panel and FASTKD2 in cyan. DAPI is shown in gray. Boxes in MERGE panels represent areas enlarged. Zoom-ins show RNA staining in gray. Arrowheads mark mitochondrial RNA granules. Scale bars in whole-cell panels represent 10 μm and in zoom-ins represent 2 μm.

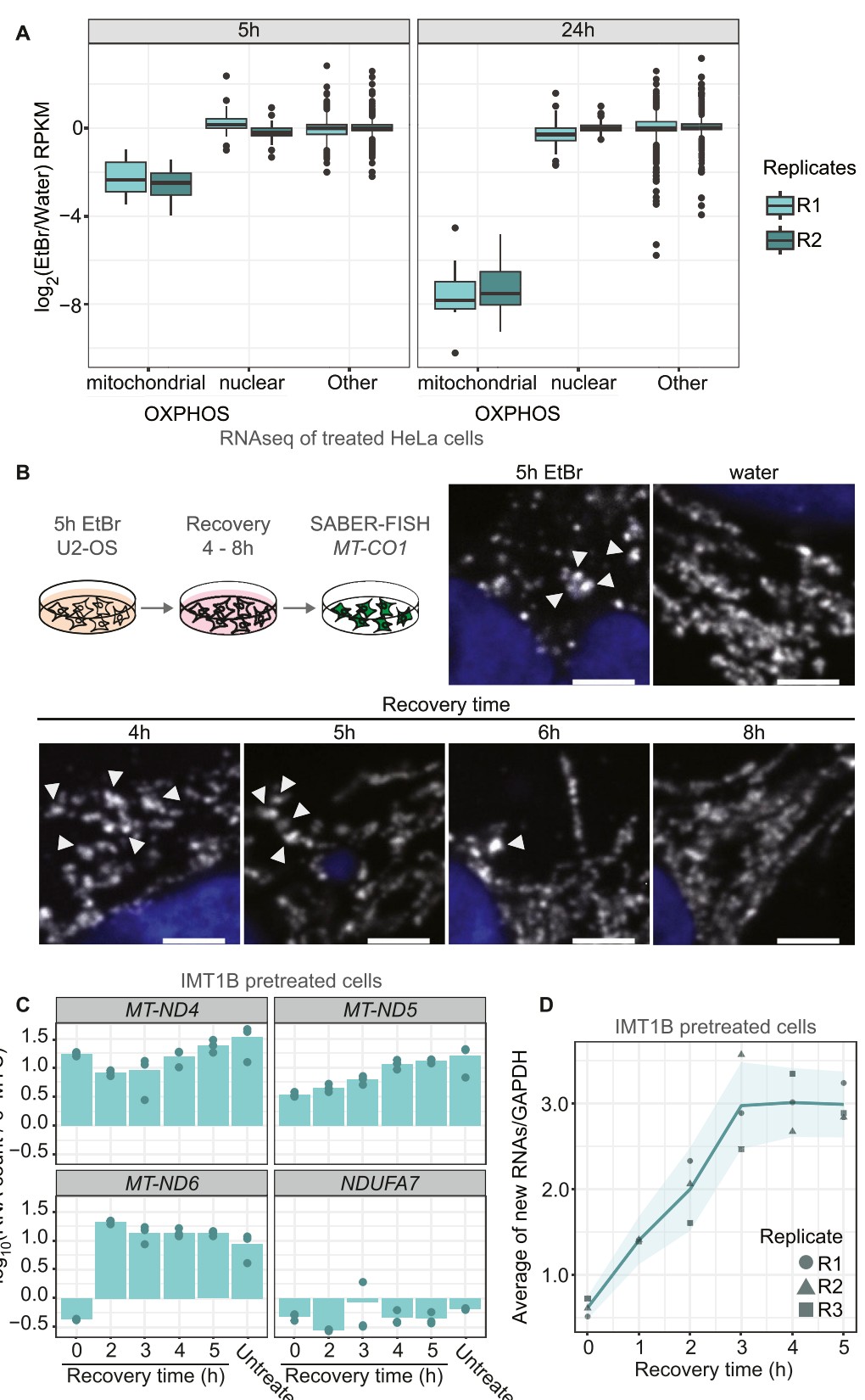

RNAseq of treated HeLa cells

IMT1B pretreated cells

### Degradation kinetics of mt-mRNAs suggest a protective function of inhibition granules

We reasoned that inhibition granules may form as a reservoir to overcome OXPHOS imbalance during stress recovery. Consistent with granules serving a protective role, RNAs remained visible after 24 h of drug treatments, despite the short half-lives of mt-mRNA at the steady state (Figs 5A and S2B). To obtain a quantitative view of mt-mRNA decay after transcription inhibition, we measured RNA abundance through time after transcription inhibition by either EtBr or IMT1B to measure half-lives at early and late time points. If inhibition granules are indeed protective, stabilization would be observed in the kinetics of RNA decay as a two-state model, where one population (or state) is degraded fast and one stays more stable. For cells treated with EtBr, we detected a two-state decay curve for *MT-ND5* and *MT-ND6*. In contrast, the nuclear-encoded control transcript *NDUFA7* had a decay curve with a one-state model (Figs 5B and S8A, Table 1). For cells treated with IMT1B treatment, we were able to detect more transcripts with decay curves consistent with a two-state model (*MT-ND1*, *MT-ND2*, *MT-ND3*, *MT-ND5*, and *MT-ND6*) (Figs 5C and S8B). Thus, we suggest that inhibition granules might play a protective role for at least some transcripts. Even though stabilization is observed with these long-term treatments, the NanoStrings data for the change in 5 and 24 h of EtBr treatment show a high correlation with the changes observed in our RNA-sequencing data (Fig S7C). In addition, we compared the early half-lives of transcripts with their mean change in the fraction of cellular area occupied and did not observe a relationship, indicating that the ability to form inhibition granules seems to be independent of transcripts' half-lives (Fig 5D).

To elucidate the fate of RNAs within inhibition granules during stress release, we employed a metabolic labeling approach. After treating cells with IMT1B for 5 h to induce inhibition granules, we removed the inhibitor and immediately added 4sU to label all newly synthesized RNAs (Fig 5E and F). This strategy allowed us to distinguish between preexisting and newly transcribed RNAs. We then measured the levels of unlabeled RNAs, which were predominantly those that had been sequestered in inhibition granules. Our analysis revealed a gradual decrease in the abundance of these preexisting RNAs, beginning ~2 h after stress release (Figs 5E and F and S8C). This decline, although slow, indicates that the RNAs are being degraded. The onset of RNA degradation coincides with the timing of inhibition granule dissolution, suggesting that mt-mRNAs lose their protective environment as the granules dissipate, becoming susceptible to degradation processes.

# Discussion

In this study, we have identified a novel type of MRG that forms in response to transcription inhibition. Our experiments employed strong perturbations to induce widespread transcription arrest across the mitochondrial network, resulting in the formation of large, easily visualized granules that persisted for hours. However, we hypothesize that smaller, transient granules may form under physiological conditions as individual mtDNA molecules switch between transcriptionally active and inactive states. During transcription arrest, we observed an imbalance in OXPHOS expression, coinciding with the sequestration of RNAs into these inhibition granules. We propose that these structures act as RNA reservoirs, facilitating recovery after stress release or upon transcription reactivation (Fig 5G).

The formation of these granules likely involves various molecular interactions, including protein–protein, RNA–protein, and RNA–RNA associations (Treeck et al, 2018; Matheny et al, 2021). Mitochondrial RNA-binding proteins, particularly members of the FASTK family known for their roles in RNA stability and processing, are potential candidates for mediating inhibition granule assembly (Jourdain et al, 2015; Popow et al, 2015; Boehm et al, 2017; Ohkubo et al, 2021).

Our data show that some RNAs become more stable during long-term transcription inhibition, contemporaneously as granules form. The mechanisms and potential function of granule formation and RNA stabilization will be the subject of future studies. It will be important to determine whether RNAs are accessible for translation in these granules, as it has been suggested for stress granules in the cytosol in the past (Mateju et al, 2020). Inhibition granules could represent an intermediate state, where RNAs are kept before transitioning onto mitochondrial ribosomes for translation. MT-RNAs are bound via the SLIRP-LRPPRC complex for transfer onto ribosomes (Sasarman et al, 2010; Ruzzenente et al, 2012; Lagouge et al, 2015; Singh et al, 2024). Thus, an intermediate might show accumulation of LRPPRC in foci. By immunofluorescence, we did not observe such an LRPPRC accumulation. Also, the inhibition of translation by chloramphenicol did not lead to granule formation. Nevertheless, in future work it will be interesting to determine

---

**Figure 4. Cells recover from transcription arrest.**
**(A)** RNA-seq after EtBr or water treatment for 5 or 24 h of HeLa S3 cells. Boxplots represent the distribution change for mitochondrial-encoded, nuclear-encoded oxidative phosphorylation subunits and other (residual transcriptome) transcripts for two biological replicates. The y-axis shows the log$_2$ change of EtBr over the water control in RPKM. **(B)** Confocal microscopy of *MT-CO1* distribution changes in the mitochondrial network during recovery. U2-OS cells were pretreated for 5 h with EtBr, washed, and grown in normal media for 4 up to 8 h. SABER-FISH for *MT-CO1* was performed, and the cells were imaged. Shown is SABER-FISH after 5 h of EtBr and water treatment as controls. Zoom-ins are shown for the different recovery time points and for the control panels. Whole cells can be found in Fig S7A. Scale bars represent 2 $\mu$m. White arrowheads mark RNA granules. All images in this figure were adjusted to represent the spatial distribution and not differences in intensities. **(C)** Relative RNA counts of cells during recovery of IMT1B-treated cells. Cells were pretreated with 10 $\mu$M IMT1B, washed, and grown in normal media. In the last 10 min of each time point, 4sU was added. RNA levels were measured after 0, 2, 3, 4, and 5 h of recovery by MitoStrings and their counts normalized to RNA counts of c-MYC. The y-axis shows the log$_{10}$ scale of relative RNA counts of either *MT-ND4*, *MT-ND5*, *MT-ND6*, or *NDUF7A* as a cytosolic control. Shown is the mean of three biological replicates, and points represent the individual replicates. **(D)** Metagene plot of transient transcription analysis during recovery. 4sU-labeled RNAs were biotinylated and enriched on streptavidin beads. Eluted RNAs were then measured by MitoStrings. Shown are the averaged relative counts of all 4sU-labeled mt-mRNAs normalized to 4sU-labeled *GAPDH* on the y-axis. The x-axis shows the time course of 0, 1, 2, 3, 4, and 5 h. Shown is the mean of the average of three biological replicates in dark green. In dark gray green and in different shapes are the single replicates plotted to show the variation of the experiment. The turquoise ribbon represents the 95% confidence interval.

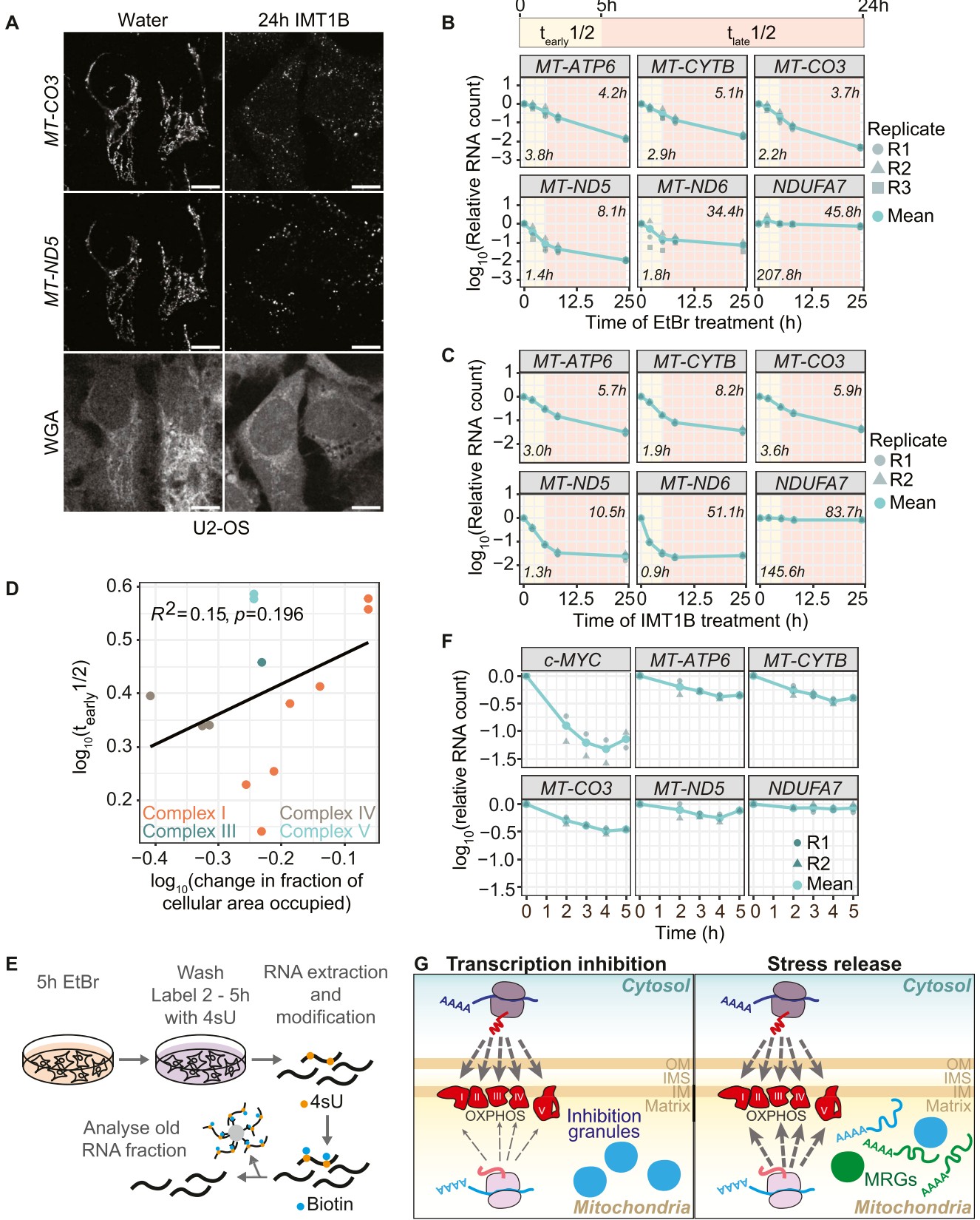

whether inhibition granules represent the stabilization of an intermediate state between RNA production and translation.

Other open questions surround the content of inhibition granules. Are they merely containing mRNAs or different RNA species? Interestingly, the noncoding antisense RNA of *MT-CO1*, *MT-ncCO1* was found to localize to these granules, indicating they contain more than mRNA. Inhibition granules might contain more than just mRNAs, and additional antisense RNAs might be present in these granules, similar to D-foci.

The discovery of inhibition granules adds to the growing list of MRGs and underscores the complexity of mitochondrial gene expression regulation. These findings reveal an additional layer of compartmentalization within mitochondrial gene expression that together with other MRGs provides multiple intervention points throughout the gene expression process, allowing cells to maintain proteostasis in the face of aberrant OXPHOS expression.

# Materials and Methods

## Cell culture

U2-OS cells (ATCC) were cultured in McCoy's 5A media (#30-2007; ATCC) supplemented with 10% FBS. HeLa cells were cultured in DMEM supplemented with 10% FBS. For cell passaging, cells were rinsed with 1x PBS and treated with Gibco TrypLE Express Enzyme (12605010; Gibco) for 15 min (U2-OS cells) or 5 min (HeLa cells). Cells were seeded onto plates with fresh media.

## Drug treatments of cells

Transcription was inhibited with either 2 µg/ml ethidium bromide, 10 µM IMT1B, or 100 µM 2′-CMA. The drugs were diluted to the respective concentration in medium and added to the cells for the indicated time points. As controls, 1% water was used for the ethidium bromide treatment and 0.1% DMSO for IMT1B and 2′-CMA. 100 µM antimycin A in EtOH or 100 µM bortezomib in DMSO was used in Fig 1D. As controls, cells were treated with 2% EtOH or 2% DMSO. Treatment time was 5 h. For treatments of cells with 100 nM antimycin A or 200 µg/ml chloramphenicol was performed for 24 h, 0.04% EtOH served as a control (Fig S1C).

## siRNA treatment against SUV3

siGENOME Human *SUPV3L1* (SUV3) siRNA was ordered in SMARTpool format from Horizon. For siRNA treatment, cells were grown until 70% confluency. Afterward, they were transfected using the Lipofectamine RNAiMAX kit (#13778075; Invitrogen). Therefore, 9 µl Lipofectamine was premixed with 350 µl Opti-MEM (#31985062; Gibco). 1.5 µl of 20 µM siRNA pools (*siSUV3* and untargeted as a control) was mixed with 350 µl Opti-MEM. Both mixtures were combined and incubated for 5 min. In the meantime, cells were supplemented with fresh media. 250 µl of the mixed siRNAs was added to the respective wells. After 24 h of siRNA treatment, the media were replaced with fresh media lacking siRNA. After an additional 48 h, cells were harvested. Some cells were seeded into glass chambers to grow overnight and could be used for SABER-FISH. A fraction of cells were frozen down to be used for Western blot analysis.

## Probe design

RNA-FISH probes were designed according to Beliveau et al (2018). Because the mitochondrial genes overlap with pseudogenes, the probe candidates were aligned against the mitochondrial DNA alone. If the whole genome was used, no specific probes could be selected, because of pseudogenes in the nuclear DNA. No K-mer filtering was used. Where necessary, probe sequences were converted to the reverse complement sequence.

After OligoMiner filtered suitable oligo sequences, they were manually extended by primer sequences to be able to do

**Figure 5. Inhibition granules show protectivity for parts of mt-mRNAs.**
**(A)** SABER-FISH for *MT-CO3* and *MT-ND5* after 24 h of IMT1B or DMSO treatment in U2-OS cells. RNAs are shown in gray, as well as WGA staining of cells. Scale bars represent 10 µm. Signals were adjusted to represent the spatial distribution and not for intensity comparisons. **(B)** Degradation kinetics of representative mt-mRNAs in EtBr long-term treatment. Cells were incubated for 0 up to 24 h in EtBr media. RNAs were extracted, and RNA abundance was measured by MitoStrings. The y-axis shows the $\log_{10}$ of the RNA counts normalized to the counts of *GAPDH*. The x-axis shows the treatment time in hours. The bar underneath represents the time frames used to calculate half-lives for early ($t_{early}1/2$) and late time points ($t_{late}1/2$). Shown is the mean of three biological replicates in turquoise. In dark green and different shapes, the single biological replicates are shown. The numbers in the left lower corner represent early half-lives and in the upper right corner the late half-lives. **(C)** Degradation kinetics of representative mt-mRNAs in IMT1B long-term treatment. Cells were incubated for 0 up to 24 h in IMT1B media. RNAs were extracted, and RNA abundance was measured by MitoStrings. The y-axis shows the $\log_{10}$ of the RNA counts normalized to the counts of *GAPDH*. The x-axis shows the treatment time in hours. The bar underneath represents the time frames used to calculate half-lives for early ($t_{early}1/2$) and late time points ($t_{late}1/2$). Shown is the mean of two biological replicates in turquoise. In dark green and different shapes, the single biological replicates are shown. The numbers in the left lower corner represent early half-lives and in the upper right corner the late half-lives. **(D)** Comparison of the early half-lives calculated for the EtBr-treated cells with the change of the fraction of the cellular area occupied. The y-axis shows the half-lives in the $\log_{10}$ space. The x-axis shows the mean of the change in the fraction of the cellular area occupied of the two replicates in the $\log_{10}$ space. The change was calculated by dividing the median of the fraction of the cellular area occupied after 5 h of EtBr treatment by the median of the water control. The single data are shown in Fig 2D. The subunits were colored according to their complex, subunits of Complex I in orange, of Complex III in dark green, of Complex IV in brown, and of Complex V in turquoise. Pearson's correlation coefficient $R^2$ was used. **(E)** MitoStrings analysis of representative unlabeled granular RNAs after stress release. Cells were treated for 5 h with EtBr, washed, and grown in media supplemented with 4sU for 0, 2, 3, 4, and 5 h. RNAs were extracted, and 4sU (shown in orange)-labeled RNA was biotinylated (biotin is shown in blue). Biotinylated RNA was bound onto streptavidin beads, and the flow-through, representing the unlabeled RNA fraction, was used for MitoStrings. **(F)** Graph showing decrease in unlabeled RNAs after stress release. The y-axis shows the $\log_{10}$ of RNA levels normalized to an unlabeled spike-in, added during RNA extraction. **(G)** During transcription inhibition, mitochondrial RNAs form inhibition granules and the oxidative phosphorylation (OXPHOS) expression inside mitochondria is decreased, whereas the nuclear expression of OXPHOS subunits remains unchanged. During stress release, OXPHOS expression will eventually increase again. Inhibition granules might help in recovery through release of stored RNAs. Mitochondrial RNA granules will form again and replenish the mitochondrial network with RNAs. Inhibition granules are shown in blue and mitochondrial RNA granules in green.

**Table 1.** Half-lives of long-term treated RNAs.

| | EtBr | | IMT1B | |
|---|---|---|---|---|
| Transcript | $t_{early}$1/2 | $t_{late}$1/2 | $t_{early}$1/2 | $t_{late}$1/2 |
| MT-ATP6 | 3.92 | 4.19 | 3.03 | 5.65 |
| MT-ATP8 | 4.32 | 4.34 | 2.89 | 5.41 |
| MT-CO1 | 2.40 | 4.92 | 3.60 | 8.62 |
| MT-CO2 | 2.31 | 4.73 | 2.94 | 6.84 |
| MT-CO3 | 2.29 | 3.68 | 3.61 | 5.93 |
| MT-CYB | 2.86 | 5.14 | 1.94 | 8.24 |
| MT-ND1 | 2.40 | 6.99 | 1.29 | 10.15 |
| MT-ND2 | 2.50 | 7.09 | 1.14 | 13.43 |
| MT-ND3 | 1.77 | 5.07 | 1.79 | 7.59 |
| MT-ND4 | 4.96 | 4.61 | 3.16 | 6.64 |
| MT-ND4L | 3.72 | 4.80 | 2.61 | 6.06 |
| MT-ND5 | 1.39 | 8.10 | 1.34 | 10.46 |
| MT-ND6 | 1.84 | 34.40 | 0.91 | 51.05 |
| NDUFA7 | *207.81 | 45.83 | 145.62 | 83.55 |
| C-MYC | −4.47 | 21.72 | −5.17 | 13.37 |

* = only 0 and 5 h were used for linear regression as the half-life resulted otherwise in negative values.

nucleotide exchange reactions with hairpins to synthesize the final oligos. All oligo sequences are listed in Table S1. Hairpin sequences were taken from Kishi et al (2019). For bicistronic transcripts (*MT-ATP6/8*, *MT-ND4/L*), probe pools were designed that detect the full bi-cistronic transcripts. All sequences used for SABER-FISH can be found in Table S1.

## Probe synthesis

Probe synthesis was performed as published previously (Kishi et al, 2019). A reaction mix without oligos was prepared on ice and incubated for 15 min at 37°C to remove any free guanines in the mix. One reaction mix contained 0.5 $\mu$M (HP27 and HP28)–1 $\mu$M hairpins (all other hairpins), 1x PBS buffer, 300 nM dNTPs without guanines, 100 nM Clean.G oligo, 10 mM MgSO$_4$, and 8 U of Bst LF polymerase (#M1213-200; BioVision) or 100 U Bst LF polymerase (#BPL-200; McLab). After the 15-min preincubation, 1 $\mu$M of oligo pool mix was added. The reaction was incubated for 80 min at 37°C followed by 20 min at 80°C to inactivate the polymerase and cooled down to 4°C. Next, the elongation was tested on 1.3% agarose gels. If an elongation of at least 100 bp was observed, probes were purified using the MinElute PCR kit from Qiagen. The ssDNA probes were eluted using 30 $\mu$l of RNase-free water, and their concentration was measured by NanoDrop.

## Coating of slides

150 $\mu$l of poly-L-lysine was added to each well and incubated for 10 min at RT. After three rinses with 1x PBS, slides were left to dry for 1 h and could be used for cell seeding.

## SABER-FISH

The SABER-FISH protocol was adapted from Kishi et al (2019). Cells were grown in poly-L-lysine–coated ibidi glass-bottom well chambers (#80807; ibidi) until 50–70% confluency. For drug treatments, cells were grown overnight in normal media and at the next day treated as described in the drug treatment section. After the treatment, cells were rinsed 2x with 1x PBS and fixed with 4% freshly prepared formaldehyde (Pierce 16% Formaldehyde [wt/vol], methanol free, #28906; Sigma-Aldrich) for 1 h at RT. Fixation time for Figs 1B and D, S1A–C, and S6A and B has been 20 min. After fixation, cells were rinsed 3x with 1x PBS. Cells would be either stored at 4°C overnight to be processed the next day or directly subjected to permeabilization. In case of SABER-FISH data used for quantification, cells were subjected to WGA staining right before permeabilization. Therefore, cells were treated with 5 $\mu$g/ml WGA-CF405S (#29027-1; Biotium) for 30 min in the dark at RT. The cells were washed three times with 1xPBS and could be permeabilized afterward. To this end, cells were incubated in 1x PBS and 0.1% TX-100 for 20 min at RT. Afterward, cells were rinsed 2x with 1x PBS, followed by a 5-min wash in 1x PBS at RT. A wash for 5 min in 1x PBS and 0.1% Tween-20 was followed. Cells were equilibrated for 15 min at 43°C in prewarmed Whyb LD buffer (2xSSC, 40% formamide, 0.1% Tween-20). During this incubation, the hybridization mix was prepared. 1 $\mu$g of each probe was mixed with the 1.4x hybridization buffer (2.8x SSC, 56% formamide, 0.14% Tween-20, 14% dextran). The ratio of hybridization buffer to water including probes was 85–35 $\mu$l per well. The mix was prewarmed to 43°C, and the Whyb was replaced with 120 $\mu$l of the hybridization mix. The cells were incubated at 43°C overnight. The next day, cells were first quickly rinsed with prewarmed Whyb and washed two times for 30 min at 43°C with Whyb buffer. Two washes at 43°C for 5 min with 2x SSC and 0.1% Tween-20 were followed. Cells were rinsed for 1 min with 1x PBS buffer at RT. They were kept in fresh 1x PBS buffer until the hybridization oven cooled down to 37°C. In the meantime, the imager probes were mixed for fluorescence detection. From here, all incubations were done in the dark to avoid bleaching of the fluorophores. The imager mix contained 1 $\mu$M of each imager probe and 1x PBS. Cells were incubated either for 30 or 60 min with imagers at 37°C. The cells were washed with prewarmed 1x PBS once for 5 min and two times for 2 min. Except for WGA-stained samples, DAPI staining was followed. Therefore, cells were rinsed once with 1x PBS at RT and incubated for 10 min in DAPI (5 $\mu$g/ml in 1xPBS and 20 U/$\mu$l of SUPER RNasin) at RT, and washed three times for 5 min with 1x PBS. 150 $\mu$l mounting media (80% glycerol, 1xPBS, 20 mM Tris, pH 8, 2.5 mg/ml n-propyl gallate, 20 U/$\mu$l SUPER RNasin) were added, and the cells were imaged either the same or the next day. For Figs 1A and D and S1A–C, the Hyb buffer and Whyb contained 1% Tween-20.

## Immunofluorescence

For immunofluorescence after SABER-FISH treatments, mounting media were removed by two quick rinses with PBST buffer (1xPBS, 0.1% TX-100). Cells were washed three to four times for 5 min with PBST followed by three washes of 5 min with displacement buffer (50% formamide, 1xPBS) to remove fluorescent probes. Next, cells

were washed three times for 2 min with PBST and rinsed three times with 1xPBS. Cells were blocked for 1 h at room temperature with blocking buffer (5% goat serum in 1xPBS and 0.1% TX-100). Cells were incubated overnight at 4°C with the primary antibodies. Unbound antibodies were removed by three washes for 5 min with PBST at RT. Cells were incubated with a secondary antibody harboring the fluorophores diluted in blocking buffer for 1 h at RT and washed again three times for 5 min with PBST at RT. After staining nuclei by DAPI, cells were covered with mounting media and were stored at 4°C until imaging.

For immunofluorescence directly after drug treatments, cells were fixed and permeabilized as described in the SABER-FISH section. Afterward, cells were incubated with blocking buffer and treated as outlined above.

## Antibodies

Listed are antibodies used in this study with information of dilution and where they are purchased from. All antibodies were diluted in blocking buffer, if not stated otherwise: TOM20 (#sc-17764; Santa Cruz) used in a 1:200 to 1:500 dilution, GRSF1 (#ab205531; Abcam) used in a 1:500 to 1:1,000 dilution, FASTKD2 (#17464-1-AP; ProteinTech) used in a 1:500 dilution, ds/ssDNA (#61014; PROGEN) used in a 1:100 dilution, LRPPRC/GP130 (#ab97505; Abcam) used in a 1:200 dilution, SUV3L (#ab127909; Abcam) used in a 1:1,000 dilution in milk for Western blot, ACTB (#3700; Cell Signaling) used in a 1:5,000 dilution in milk for Western blots, HRP-conjugated anti-rabbit IgG (cat. no. 7074S; Cell Signaling) used in a 1:10,000 dilution in 5% milk, HRP-conjugated anti-mouse IgG (cat. no. 70765; Cell Signaling) used in a 1:10,000 dilution in 5% milk, goat anti-rabbit IgG Alexa Fluor 647 (cat. no. A-21245; Thermo Fisher Scientific) used in a 1:1,000 dilution, goat anti-mouse IgG2a Alexa Fluor 488 (cat. no. A-21131; Thermo Fisher Scientific) used in a 1:1,000 dilution, and goat anti-mouse IgM Alexa Fluor 555 (cat. no. A-21426; Thermo Fisher Scientific) used in a 1:1,000 dilution.

## Western blotting

Cells were lysed in RIPA buffer and treated with Benzonase for 30 min and afterward mixed with loading buffer (95 mM Tris–HCl, pH 6.8, 7.5% glycerol, 2% SDS, 0.5 mg/ml bromophenol blue, 50 mM DTT). Proteins were separated on 10% NuPAGE Novex Bis-Tris gels (Thermo Fisher Scientific) and transferred to a nitrocellulose membrane. Membranes were incubated with primary antibodies overnight (SUV3L and ACTB). For visualization, membranes were incubated with secondary antibodies conjugated with horseradish peroxidase for chemiluminescence detection.

## Imaging

Samples were imaged on a Nikon Ti inverted microscope with a W1 Yokogawa Spinning disk with 50-$\mu$m pinhole disk, a Nikon motorized stage, a Physik Instrument Piezo Z motor, an Andor Zyla 4.2-Plus sCMOS monochrome camera VSC-06522 or VSC-10150 (used for quantification images of two replicates of IMT1B treatment and *MT-ND5* of EtBr treatment), and OKO Lab Heated enclosure with $CO_2$ control at room temperature. For imaging, Nikon Elements

Acquisition Software AR 5.02 was used. All files were saved in .ND2 format. Before each imaging, slides were cleaned first with Sparkle and water to remove salts and finally with 70% EtOH to remove any oils. Samples for quantifications were imaged with a Plan Apo $\lambda$ 60x/1.4 Oil DIC objective in combination with immersion oil (Cargille, nondrying immersion oil Type 37, #16237) and with a laser device of Digital Mirror Device lines (Aura/LED) 350, 380, 488, and 561 with emission filter 455/50, 480/40, 525/36, 605/52, 630/75, and 705/72 or with laser lines 405, 488, 561, 640 (Nikon Instruments, Laser Unit model Lun-F) and emission filters (455/50, 480/40, 525/36, 605/52, 630/75, 705/72). For each image set, 10 fields of view of an empty ibidi glass chamber well were imaged with the same laser settings used for sample acquisition for flat-field correction. Z-stacks of 37 planes with a size of 200 nm were imaged, with no binning and dual-gain ¼ setting. For all images, z-stacks were imaged in the following order: first, all laser lines were imaged before the next z-plane. The final pixel size was 0.108 $\mu$m × 0.108 $\mu$m. Exposure times (usually between 500 and 700 ms) and laser power varied between experiments because of different probe sets used in SABER-FISH. If not stated differently, images of the residual experiment were taken with no binning and dual-gain ¼ settings, and using a silicon oil objective with correction collar (Plan Apo $\lambda$S SR HP 100xC/1.45 Silicon DIC) in combination with silicon immersion oil 30cc (#MXA22179; Nikon). For collar correction of the silicon oil objective, fluorescent beads were prepared as described in Isaac et al (2024). Z-stacks with varying plain number-to-image full volumes of cells and sizes of 200 nm or 300 nm were taken. The final pixel size was 0.065 $\mu$m × 0.065 $\mu$m. RNase- and DNase-treated samples were imaged with a Plan Fluor 40x Oil DIC H/N2 objective in combination with immersion oil, 12-bit, and no binning settings. The final pixel size was 332.8 $\mu$m × 332.8 $\mu$m.

## Image analysis

### *Flat-field correction of images*
For image analysis, the images were first background-corrected in Fiji. Therefore, the average of 10 fields of view of an empty well imaged with the same microscopic settings used to image the samples of each experiment were averaged. The sample image and the background image were corrected for the camera error by subtraction. Afterward, the sample image was divided by the background. Next, we followed a z-projection of the z-stack using the sum of each slice in Fiji.

### *Segmentation of whole cells and RNAs*
Segmentation of cells and RNA signal was done in CellProfiler. For whole-cell segmentation, the whole range of intensities was chosen and the cells were segmented by hand based on the WGA staining. To segment the RNA signal, again the full intensity range was used. Next, the borders were enhanced using the Kirsch filter. To detect primary objects, a global threshold was used with the following settings: minimum cross-entropy, threshold smoothing scale of 1.3488, threshold correction factor of 1, lower and upper bounds were set to 0 and 1, no log transformation was done before thresholding and intensities were used to distinguish between clumped objects and used to draw dividing lines. The size used for the smoothing filter for declumping, as well as the minimum

allowed distance between maxima, was set to automatic calculation. Full resolution was used to determine local maxima. Holes in identified objects were filled after declumping.

The segmented RNAs were assigned to the segmented cells, and the area and the intensities for the segmented cells and the RNAs were extracted. Intensity extraction was performed based on the RNA channel. The data could then be analyzed in R. The extracted data of the segmentation, used for analysis, as well as normalized data, are shown in Tables S2A and B and S3.

### Calculating the cellular area occupied

To calculate the cellular area occupied, the area of each RNA object segmented of a cell was summed up and divided by the area of the parent cell. The distribution was plotted as a swarm plot to represent single cells, and on top of those, boxplots were plotted to show the median. To test if there was a significant change, a one-sided $t$ test was performed using the rstatix R package and the following settings: pairwise_t_test (NormArea ~ Time, paired = FALSE, alternative = "less," p.adjust.method = "none"), where NormArea is the fraction of cellular area occupied. Statistical results are summarized in Table S4.

### Single-cell intensity analysis

For intensity analysis of single cells, the integrated intensities measured of each RNA object in a single cell were summed and normalized to the cell area. Again, single-cell data were blotted as swarm plots and boxplots.

### Measurement of skewness of the signal distribution

Masks for single cells were converted in Fiji to ROIs, and the skewness measurement of Fiji was used to extract data from the single RNA channels of the background-corrected z-projections. The data were analyzed in R and plotted as swarm and boxplots. The extracted data can be found in Table S5. Significance of the change was tested with a one-sided $t$ test using the rstatix R package and the following settings: pairwise_t_test (Skew ~ Treat, paired = FALSE, alternative = "greater," p.adjust.method = "none"). Statistical results are summarized in Table S4.

### Image manipulations

To represent images in figures, brightness and contrast were adjusted for the whole fields of view in Fiji to represent the full spatial distribution of RNAs. This was necessary because of high differences in intensities between control and sample and in an image itself. Thus, images do not represent differences in intensities. Images were then cropped for single cells and zoom-ins using regions of interest. For figures showing WGA staining, a mean filter size 1 was used for the WGA channel. All other channels were only adjusted for brightness and contrast. TEM images were just cropped to show regions of interest of different cells.

### TEM

Cells were treated for 24 h with IMT1B, DMSO or for 5 h with EtBr. Cells were fixed in 2.5% glutaraldehyde, 1.25% PFA, and 0.03% picric acid in 0.1 M sodium cacodylate buffer, pH 7.4, for 1 h. The cells were then postfixed for 30 min in 1% osmium tetroxide (OsO4)/1.5% potassium ferrocyanide (KFeCN6), washed in water 3x, and incubated in 1% aqueous uranyl acetate for 30 min followed by 2 washes in water and subsequent dehydration in grades of alcohol (5 min each; 50%, 70%, 95%, 2 × 100%). Cells were removed from the dish in propylene oxide, pelleted at 7,000$g$ for 3 min, and infiltrated for 2 h ON in a 1:1 mixture of propylene oxide and TAAB Epon (TAAB Laboratories Equipment Ltd, https://taab.co.uk). The samples were subsequently embedded in TAAB Epon and polymerized at 60°C for 48 h.

Ultrathin sections (about 80 nm) were cut on a Reichert Ultracut-S microtome, picked up onto copper grids stained with lead citrate, and examined in a JEOL 1200EX transmission electron microscope or a Tecnai G2 Spirit BioTWIN, and images were recorded with an AMT 2k CCD camera.

### Quantifying mitochondrial phenotypes of TEM data

Mitochondria of 10 images for EtBr and IMT1B samples and 11 images of DMSO were quantified by hand by two people in a double-blind fashion, and the data were analyzed in R.

### qRT-PCR to measure RNA abundance after EtBr treatment

HeLa S3 cells were grown until 65–70% confluency, washed with prewarmed 1xPBS, and treated for 5 h with 0, 0.5, 1, or 2 $\mu$g/ml ethidium bromide for 5 h. Cells were scraped, spun down, and finally lysed in 750 $\mu$l Quizol and frozen for later use. RNA was isolated as described below. The RNA still containing DNA was resuspended in 30 $\mu$l water. To remove DNA, DNA-*free* DNA Removal Kit from Thermo Fisher Scientific (cat. no. AM1906) was used. In brief, samples were treated with DNase I for 30 min at 37°C and the DNase was inactivated via the addition of the DNase inactivation reagent. Samples were centrifuged at 10,000$g$ for 90 s, and the supernatant was transferred to a new tube. The SuperScript III First-Strand Synthesis System from Thermo Fisher Scientific (cat. no. 18080051) was used to synthesize cDNA with the addition of random hexamers according to the instructions. cDNA was then used for qRT-PCR. 1 ng/$\mu$l cDNA was used per reaction. cDNA was mixed with the SsoFast EvaGreen Supermix from Bio-Rad (cat. no. 1725201) and 0.4 $\mu$M primers (MT-CO1 For: 5'-CTATGA-TATCAATTGGCTTCCTAG-3', MT-CO1 Rev: 5'-TGGTAGCGGAGGTGAAA-TATGCTCGTGTGT-3', MT-ND1 For: 5'-TGCCATCATGACCCTTGGCCAT-3', MT-ND1 Rev: 5'-GCCTGAGACTAGTTCGGACTCCCCTTCGGC-3', NDUFA7 For: 5'-CATGACCTGCAGGGGAAGCT-3', NDUF7A Rev: 5'-CCA-CAGGGAGCTTGGGAGGA-3', GAPDH For: 5'-GTCTTCACCACCATGGA-GAAGG-3', and GAPDH Rev: 5'-ATGATCTTGAGGCTGTTGTCAT-3'). GAPDH primer sequences were taken by Nagao et al (2008). The qRT-PCR was started in Bio-Rad CFX384 Touch Real-Time PCR Detection System. The following program was used: 30 s at 95°C followed by 40 cycles of 95°C for 5 s, 55°C for 10 s, and a melting curve starting at 65°C until 95°C in 0.5°C for 5 s.

### Transient transcription analysis

To measure transient transcription during recovery, cells were first treated for 5 h with IMT1B as described above. The cells were

washed three times with 1xPBS and grown in normal media for recovery. To label nascent RNAs, 4-thiouridine (4sU, T4509; Sigma-Aldrich) was directly added to the cells to a final concentration of 500 µM at the last 10 min of each time point. Samples were harvested after 0, 1, 2, 3, 4, and 5 h of recovery. Therefore, the media were removed and cells placed on ice. They were scraped directly into Quizol, and the lysates were transferred into tubes. 2 mM DTT was added to each tube, and the samples were heated for 3 min at 65°C and stored at –80°C for RNA extraction, followed by biotinylation and enrichment of labeled RNAs to be analyzed by MitoStrings.

## 4sU labeling to distinguish between old and new RNAs during recovery

To measure RNA abundance of old RNAs after stress release, cells were first treated with EtBr for 5 h as described above. Afterward, cells were washed three times with 1x PBS. Cells were then incubated in media supplemented with 500 µM 4sU for 2, 3, 4, or 5 h. For time point 0 h, cells were harvested directly after EtBr treatment. Harvesting of cells was performed as described in the transient transcription section. The lysates could be used for RNA extraction and biotinylation. The labeled RNAs were bound onto streptavidin beads, and the flow-through fraction was used for analysis by MitoStrings.

## RNA extraction

RNAs of long-term treated cells and for RNA-seq were extracted as follows. After the respective treatment times, plates were put on ice. Cells were washed with ice-cold 1xPBS and scraped into Quizol. Lysates were stored at –80°C until RNA extraction. Lysates were thawed at RT. 200 µl chloroform was added to 1 ml of sample, mixed by vortexing, and incubated for 2 min at RT. Lysates were centrifuged for 15 min at 4°C and at 12,000$g$. The aqueous phase was transferred into a fresh tube without disturbing the interphase. RNAs were precipitated with 500 µl isopropanol; in case of qRT-PCR samples, 1 µl Invitrogen GlycoBlue from Invitrogen (AM9515) was added, and incubated either at RT for 10 min or at –80°C overnight. After a second spin at 4°C and 12,000$g$ for 10 min, the supernatant was discarded. The pellet was washed with 75% EtOH and pelleted again by centrifugation at 4°C and 7,500$g$ for 5 min. The supernatant was discarded and the pellet dried for 5–10 min at RT. The RNAs were resuspended in 32 µl of RNase-free water, and the concentration was measured by Qubit and NanoDrop. 40 ng of RNAs was used in MitoStrings measurements. For samples used to study the fate of old RNAs after stress release, 1 ng/µl unlabeled in vitro synthesized ERCC-00048 spike-in was added to the thawed cell lysates. In case of transient transcription analysis, 0.25 ng/µl of in vitro synthesized 4sU labeled ERCC-00136 spike-in (as described in McShane et al [2024]) was added additionally. RNAs were extracted as described previously, and in these cases, RNA was resuspended in 50 µl of RNase-free water.

## RNA biotinylation and purification

A maximum of 40 µg of RNA per sample was used for the biotinylation reaction. Samples were brought to a volume of 70 µl and incubated at 60°C with 800 rpm in a thermoblock with rotation function for 10 min. Afterward, samples were immediately incubated for not longer than 2 min on ice. RNAs were biotinylated in a reaction containing 1x labeling buffer (20 mM Hepes, pH 7.4, 1 mM EDTA) and 0.5 mg/ml biotin-MTS (#90066; Biotium) in 20% dimethylformamide (Sigma-Aldrich). The reaction mixture was incubated in a thermoblock with rotation function at 800 rpm and 24°C for 30 min in the dark. Next, free biotin was removed. To this end, 2-ml phase-lock heavy gel tubes (#2302830; 5Prime) were prepared by centrifugation at 14,000$g$ for 1 min. The biotinylated RNA was diluted with water, transferred into the phase-lock tube, and mixed with 1 volume of chloroform:isoamyl alcohol (24:1) by manually shaking for 15 s. A centrifugation step for 5 min at RT and 16,000$g$ was followed. The upper phase was transferred into a fresh tube, and the RNA was precipitated by the addition of 0.1 volume of 5 M NaCl and 1 volume of isopropanol. The mixture was centrifuged for 40 min at 4°C and 20,000$g$. The supernatant was removed and the pellet washed with 75% EtOH, followed by a 5-min spin at 4°C and 7,500$g$. The supernatant was removed, and the pellet was dried for 5 min at RT. The RNA was resuspended in 40 µl of RNase-free water. Samples could be stored at –80°C for enrichment. The biotinylated RNA was purified with streptavidin beads of the uMACS Streptavidin kit (#130-133-282; Miltenyi Biotec). 3 µg RNA was mixed with streptavidin beads, incubated for 15 min at RT, and subsequently loaded onto a uMACS column. The flow-through was loaded again onto the columns to increase biotin binding. The flow-through contained the unlabeled RNA fraction. The beads were washed three times with prewarmed (65°C) washing buffer (100 mM Tris, pH 7.5, 10 mM EDTA, 1M NaCl, and 0.1% Tween-20) and three times with washing buffer at RT. RNAs were eluted twice by incubation at 65°C in elution buffer (550 mM Tris, pH 9, 10 mM EDTA, 150 mM TCEP). For transient transcription analysis, the elution fraction, and for the analysis of old RNA, the flow-through fraction were purified using the miRNeasy Micro kit following the manufacturer's instructions including the DNase treatment (#217084; QIAGEN). 30 ng RNA was used for MitoStrings analysis.

## MitoStrings

MitoStrings is a mitochondrial-tailored NanoStrings measurement (Wolf & Mootha, 2014). According to the manufacturer's protocol, RNAs (30–40 ng) were hybridized for 16 h at 67°C with the XT Tagset-24 (NanoString Technologies) and with DNA probes specific for mitochondrial RNA (MitoStrings probes were modified as in Isaac et al [2024]) in hybridization buffer (NanoString Technologies). Afterward, samples were loaded onto a nCounter Sprint cartridge and quantified by the use of the nCounter SPRINT Profiler (NanoString Technologies) of the Bauer core facility of Harvard or at the Manning Lab (Harvard T.H. Chan School of Public Health).

## Total RNA sequencing

### Sample collection

To analyze changes in OXPHOS expression, total RNA-seq was performed. HeLa cells were treated for 5 or 24 h with EtBr or water as described before. Cells were harvested, and the RNAs were extracted. The SMARTer Stranded Total RNA HI Mammalian kit (#634873; Takara) was used to prepare sequencing libraries according to the manufacturer's instructions. The libraries were measured on the NovaSeq (Illumina) by the Biopolymers Facility at Harvard Medical School.

### Read alignment

The provided fastq files were first concatenated to combine the data. Next, adapter sequences were trimmed as described in McShane et al (2024). The trimmed reads were filtered after mapping to rRNA using bowtie1 (Langmead et al, 2009) with settings allowing for one internal and one 5' mismatch. The residual reads were aligned to the GRCh38 human reference genome using STAR 2.7.3 (Dobin et al, 2013) with the following parameters: --outSAMtype BAM SortedByCoordinate --outReads Unmapped Fastx --outFilterIntronMotifs RemoveNoncanonical Unannotated --outFilterMultimapNmax 100.

### Expression analysis

To analyze expression changes of OXPHOS genes, read counts were normalized to length- and library size-normalized read counts. Read counts were summed over protein-coding regions with the help of the R package featureCounts (Liao et al, 2019) with the parameters: isGTFAnnotationFile = TRUE, useMetaFeatures = TRUE, countMultiMappingReads = FALSE, strandSpecific = 1, isPairedEnd = TRUE, nthreads = 4. GTF files used in this study are described here (Soto et al, 2022). Table S6 contains the RNA-seq data.

### Half-life calculations

To calculate half-lives, RNA abundance was measured in long-term transcription inhibition treatments by MitoStrings. RNA levels were measured after 0, 2, 5, 8, and 24 h of transcription inhibition. The mean of three biological replicates (EtBr-treated cells) or two biological replicates (IMT1B-treated cells) of GAPDH-normalized RNA counts was used for calculating half-lives. 0-, 2-, and 5-h measurements were used in linear regression with log transformation, and time point 0 h was set as 100%. The slope was used to calculate the half-lives: $(\log_{10}(0.5)-\log_{10}(1))/\text{slope}$. The same procedure was performed for late half-lives. Here, the 5-h measurements were set as 100%. For *NDUFA7*, we used for early half-lives only 0 and 5 h as the 2-h time point increased and led to negative half-lives. All NanoStrings data collected can be found in Table S7.

### Correlation analysis

Pearson's correlation analyses were performed using the smplot2 package in R with the following command: sm_statCorr (r2 = TRUE, color = "black," size = 0.8).

## Use of AI

ChatGPT was used to help formulate R code for half-life analysis, to extract statistics, and to write functions for image analysis. Claude was used to edit and improve the text.

## Data Availability

Sequencing data are available at GEO with the accession number GSE277754. Raw data of images used for quantification are available at OMERO (https://omero.hms.harvard.edu/webclient/?show=project-10957). The residual source images are accessible at BioImaging Archive with the accession number S-BIAD1860.

## Supplementary Information

## Acknowledgements

We thank the Churchman laboratory for helpful discussion. We thank R Stefan Isaac for helpful discussions and comments. We thank Nick Kramer for help with imaging. We thank Junior Prof. Yury Bykov and Tamara Flohr for help with TEM analysis. We are grateful to the MicRon Facility at HMS, especially Paula Montero Llopis and Praju Vikas Anekal for advice and help with imaging and analysis. We want to thank the Biopolymers Core Facility at Harvard Medical School for sequencing services, and the Boston Children's Hospital Molecular Genetics Core, the Bauer Facility at Harvard, and the Manning Lab at Harvard T.H. Chan School of Public Health for NanoStrings services. This work was supported by the NIH (R01-GM123002 to LS Churchman), a Helen Hay Whitney Foundation fellowship (F-1240 to KG Hansen), an EMBO fellowship (ALTF 762-2019 to KG Hansen), and a PROMOS fellowship to CAM Weiss.

### Author Contributions

KG Hansen: conceptualization, data curation, formal analysis, funding acquisition, validation, investigation, visualization, methodology, and writing—original draft, review, and editing.
A Baxter-Koenigs: data curation and formal analysis.
CAM Weiss: conceptualization, data curation, and formal analysis.
E McShane: conceptualization, data curation, and formal analysis.
LS Churchman: conceptualization, resources, software, supervision, funding acquisition, investigation, project administration, and writing—original draft, review, and editing.

### Conflict of Interest Statement

The authors declare that they have no conflict of interest.

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
