## [Reviewer comments · Life Science Alliance]

Life Science Alliance

Transcription arrest induces formation of RNA granules in mitochondria

Katja Hansen, Autum Baxter-Koenigs, Caroline Weiss, Erik McShane, and L. Stirling Churchman

DOI: <https://doi.org/10.26508/lsa.202403082>

Corresponding author(s): L. Stirling Churchman, Harvard Medical School

Review Timeline:

Submission Date:	2024-10-09
Editorial Decision:	2024-11-01
Revision Received:	2025-04-25
Editorial Decision:	2025-05-28
Revision Received:	2025-06-01
Accepted:	2025-06-02

Scientific Editor: Tim Fessenden

Transaction Report:

November 1, 2024

Re: Life Science Alliance manuscript #LSA-2024-03082-T

L. Stirling Stirling Churchman
Harvard Medical School
Department of Genetics
77 avenue louis pasteur
New Research Building, 356
Boston, MA 2139

Dear Dr. Churchman,

Thank you for submitting your manuscript entitled "Transcription arrest induces formation of protective RNA granules in mitochondria" to Life Science Alliance. The manuscript was assessed by expert reviewers, whose comments are appended to this letter. We invite you to submit a revised manuscript addressing the Reviewer comments.

Thank you for this interesting contribution to Life Science Alliance. We are looking forward to receiving your revised manuscript.

Sincerely,

B. MANUSCRIPT ORGANIZATION AND FORMATTING:

Reviewer #1 (Comments to the Authors (Required)):

In this paper, the authors propose that under mitochondrial transcription stress, a new type of RNA granules, termed "RNA inhibition granules," may act as reservoirs during cellular stress and subsequently resolve post-stress to facilitate the resumption of oxidative phosphorylation (OXPHOS) biogenesis. While this concept is intriguing, the study presents several limitations. Notably, the authors do not provide evidence demonstrating that translation is promptly resumed following stress relief and the release of these RNAs. Additionally, the mechanisms and factors driving the formation of inhibition granules have not been explored, resulting in a manuscript that, while interesting, comes across as somewhat descriptive

In addition to these general comments, I am listing below a few specific suggestions to improve the manuscript:

1. The authors should clarify how they identified the granules in their preliminary screening using bortezomib, AmA, EtBr, and chloramphenicol. The zoomed images suggest that the granules appear quite similar, making differentiation difficult.
2. As a necessary control, it would be beneficial for the authors to verify transcription is indeed inhibited under their experimental conditions and quantify the extent of this inhibition.
3. The authors should consider whether the mt-mRNAs within these granules are entirely unavailable for translation.
4. In describing SUV3, I recommend referring to it as "a subunit of the mitochondrial degradosome" rather than "a subunit of the mitochondrial exonuclease," as SUV3 functions as a helicase, and other exonucleases (e.g., ExoG) are present in mitochondria.
5. The authors should confirm that, under their experimental conditions, the inhibition of SUV3 leads to an increase in 7S RNA and assess the extent of transcriptional arrest.
6. The rationale behind the authors' claim that these granules protect mt-mRNA is unclear, especially since Figure 4 suggests degradation is still occurring. Although a two-stage degradation process was observed, Figure 4A shows a substantial decrease in mt-mRNA levels after 24 hours compared to 5 hours. The authors need to address this apparent discrepancy.
7. To convincingly demonstrate that the inhibition granules and mitochondrial RNA granules (MRG) are distinct structures, I recommend performing an overlay of the FASTKD2/GRSF1 markers with their FISH probes. This could be achieved through a combination of FISH and immunofluorescence or by using a fluorescently tagged marker of MRG prior to FISH.
8. The authors should explain the rationale for using NDUFA7 as the sole control. Including multiple controls would strengthen the generality of the observed effects.
9. For the TEM analysis presented in Figure EV3B, I recommend providing quantification to support claims of structural differences. A representative micrograph alone is generally insufficient, as mitochondrial ultrastructure can vary considerably.
10. The authors should investigate whether the inhibition granules contain only mRNAs or if they also include other RNA species.

Minor points:

1. The authors should clarify the rationale behind the differing formaldehyde fixation times used in various experiments.
2. NDUFA7 is misspelled in line 218.

Reviewer #2 (Comments to the Authors (Required)):

Hansen et al. investigate the spatial distribution of mitochondrial RNAs (mt-RNAs) during stress conditions using single-molecule SABER-FISH. The authors report that transcription inhibition leads to the formation of a novel form of RNA granules in mitochondria, termed "inhibition granules." They further demonstrated that cells can recover from transcription inhibition by resuming transcription and redistributing mtRNA throughout the mitochondrial network. The authors conclude that inhibition granules may function as RNA reservoirs, potentially aiding in the restoration of mitochondrial function when stress is alleviated.

The topic of mitochondrial gene expression is interesting and important, as it directly influences cellular energy production and is linked to various diseases. The manuscript is generally interesting, the experiments are well-performed, and the text is well-written. We noticed that certain controls are missing and some conclusions are not fully supported by experimental data and would propose the authors to address a few points:

Major Comments:

- The authors claim that inhibition granules have a protective function mostly based on the bimodal kinetics of mtRNA degradation following transcription inhibition. While this is interesting and possible, at this point it is only correlative and not supported by experiments. The authors should experimentally test if preventing inhibition granules formation affects RNA stability, or, if the tools are not available, temper their title and text to reflect that this is a proposed mechanism only.
- Two control experiments would strengthen the findings from the authors on the existence of inhibition granules:
 1. As an approach orthogonal to FISH, could the authors label mitochondrial transcripts to homogeneity with bromouridine (BrU), block transcription with EtBr, and look at inhibition granules after e.g. 5h with anti-BrU antibodies? This would also allow co-staining with antibodies against other intra-mitochondrial structures.
 2. The authors should test for cross-reactivity of their probe with the mtDNA nucleoids specifically in EtBr-treated cells, where the general background of free mtRNA is dissipated, and strands of the mtDNA molecule might be exposed. Could the authors repeat their SABER-FISH + DNase-treatment, like in Fig. EV1A, but in EtBr-treated cells? It will be important to use a DNase that digests both ssDNA and dsDNA.
- The effects of the treatments on RNA reorganization are inconsistent across the figures, and a better detection and quantification of the granules across some of the most important experiments would be helpful.

Minor Comments:

- Line 76: Dhir et al. (2018) do not mention the storage of dsRNA in granules.
- Depletion of SUV3, which induces the formation of dsRNA that can be stained by SABER-FISH, is not an ideal model for confirming that granule formation is triggered by transcription arrest. Downregulating POLRMT would be a better approach.
- At the level of individual mRNAs, is there a correlation between the degradation kinetics (Figure 5) and the ability of mtRNA to form clusters?

Reviewer #3 (Comments to the Authors (Required)):

The authors have produced a neat manuscript to report that on various forms of mtDNA transcription inhibition, mt-mRNA species appear to be sequestered into aggregates that are distinct from the well described mitochondrial RNA granules. The data is well presented and the paper is very interesting. All expts are performed well and data analysis is good. The authors find that for most mt-mRNA species there are essentially two differing stabilities after transcriptional inhibition. I have several questions that do not detract from the high quality of this paper but the authors may wish to address them. First, do their data really show that subsets of the mt-mRNAs are sequestered into novel granules on transcriptional inhibition? Is it not possible that all transcripts pass through this intermediate granule that is a natural step in expression? It is merely that inhibition leads to the transcripts remaining in this intermediate stage, possibly as a consequence of the stress? Could it be simply that the transcript remains associated with the stalled or incomplete mitoribosome? It would be very interesting to know whether this stress intermediate can be labelled with anti-mitoribosome antibodies. Or could the mt-mRNAs remain associated with SLRP/LRPPR prior to being transported to the mitoribosome? I guess what we're missing here is to have some idea/marker as to what this intermediate may be? Are the stored transcripts translated? Irrespective, its a very neat piece of work.

Reviewer #1 (Comments to the Authors (Required)):

In this paper, the authors propose that under mitochondrial transcription stress, a new type of RNA granules, termed "RNA inhibition granules," may act as reservoirs during cellular stress and subsequently resolve post-stress to facilitate the resumption of oxidative phosphorylation (OXPHOS) biogenesis. While this concept is intriguing, the study presents several limitations. Notably, the authors do not provide evidence demonstrating that translation is promptly resumed following stress relief and the release of these RNAs. Additionally, the mechanisms and factors driving the formation of inhibition granules have not been explored, resulting in a manuscript that, while interesting, comes across as somewhat descriptive.

Response: We thank the reviewer for the comments. We agree that our data do not provide a mechanism for the formation or function of these granules. We now make it clear in the text that the model is a hypothesis and that there are unanswered questions that remain. This manuscript complements other recent findings on RNA granule formation within the mitochondrial network (PMID: 37468471, PMID: 40249810), enriching this burgeoning field.

In addition to these general comments, I am listing below a few specific suggestions to improve the manuscript:

1. The authors should clarify how they identified the granules in their preliminary screening using bortezomib, AmA, EtBr, and chloramphenicol. The zoomed images suggest that the granules appear quite similar, making differentiation difficult.

Response: To more clearly see the granules across conditions, we have added *MT-ND2* FISH data, which better distinguishes the differences across conditions (Fig. 1D). Additionally, we have included the entire field of view in Fig. S1B. We thank the reviewer for the comment.

2. As a necessary control, it would be beneficial for the authors to verify transcription is indeed inhibited under their experimental conditions and quantify the extent of this inhibition.

Response: To determine the impact of EtBr on transcription, we took three approaches. First, we screened different concentrations of EtBr to determine when RNA abundance would be most impacted, which was the case with 2 $\mu\text{g/ml}$ EtBr (Fig. S1D). Second, we labeled RNA with 5-EU during 5 hours of EtBr treatment and performed click-chemistry to label the 5-EU-incorporated RNAs with a fluorophore. No cell shows normal RNA distribution after 5h of EtBr treatment (Figure RTR1).

96.3% with 5-EU labeling 0% with 5-EU labeling

Figure RTR1: RNA labeling with 5 μ M 5-EU for 5 hours during the treatment with water or EtBr. Cells were counted according to full network labeling.

Third, we inspected GRSF1 localization by immunostaining (Fig. 3A, C). We found that upon EtBr treatment, GRSF1 is no longer confined to granules and is distributed in the network. GRSF1 requires nascent mtRNAs to form mitochondrial RNA granules, so the loss of granule localization indicates a loss of active transcription. GRSF1 similarly loses granule localization after IMT1B treatment.

3. The authors should consider whether the mt-mRNAs within these granules are entirely unavailable for translation.

Response: While this is an important question, it is not easy to address. We now raise this point as an important open question in our discussion. *“Our data shows that some RNAs become more stable during long-term transcription inhibition, contemporaneously as granules form. The mechanisms and potential function of granule formation and RNA stabilization will be the subject of future studies. It will be important to determine if RNAs are accessible for translation in these granules, as it has been suggested for stress granules in the cytosol in the past (Mateju et al. 2020).”*

Addressing this point directly would require extensive method development that is outside the scope of this study, but is an aim of our future work.

4. In describing SUV3, I recommend referring to it as "a subunit of the mitochondrial degradosome" rather than "a subunit of the mitochondrial exonuclease," as SUV3 functions as a helicase, and other exonucleases (e.g., ExoG) are present in mitochondria.

Response: We changed the text as suggested.

5. The authors should confirm that, under their experimental conditions, the inhibition of SUV3 leads to an increase in 7S RNA and assess the extent of transcriptional arrest.

Response: To address this point, we stained for GRSF1 under these conditions, which shows that only some parts of the mitochondrial network have lost granular GRSF1 formation, an indicator for active transcription (Figure S5C). As SUV3 depletion leads to a complicated phenotype, we have removed the suggestion that the SUV3 experiment serves as an orthogonal inhibition of transcription in our study. Rather, we introduce SUV3 now as a condition where we observe RNA granules similar to the EtBr and IMT1B treatments,

consistent with its known role in transcription regulation: “Recently, it has been found that downregulation of *SUV3* leads to reduced transcription (Zhu et al. 2022). Indeed, when we knockdown *SUV3* we see that *GRSF1* shows less localization to MRGs (Fig. S5B, C). In these conditions, we also observe similar RNA granules as seen after EtBr or *IMT1B* treatment (Fig. S5D).”

6. The rationale behind the authors' claim that these granules protect mt-mRNA is unclear, especially since Figure 4 suggests degradation is still occurring. Although a two-stage degradation process was observed, Figure 4A shows a substantial decrease in mt-mRNA levels after 24 hours compared to 5 hours. The authors need to address this apparent discrepancy.

Response: Degradation is still occurring; however, it is slower, which we observe is due to a more stable subset of the RNA population. After 24 hours, a significant amount of degradation has occurred. However, the RNA-seq data agree well with the kinetic analysis we performed with NanoString. We agree that this seeming discrepancy could be confusing. To solve this problem, we now directly compare the two measures. The two data sets compare quite well ($r^2 > 0.85$; Fig. S7C).

7. To convincingly demonstrate that the inhibition granules and mitochondrial RNA granules (MRG) are distinct structures, I recommend performing an overlay of the FASTKD2/*GRSF1* markers with their FISH probes. This could be achieved through a combination of FISH and immunofluorescence or by using a fluorescently tagged marker of MRG prior to FISH.

Response: We tried many times to co-stain RNAs and proteins, however, the antibodies for *GRSF1* and FASTKD2 were not compatible with the FISH protocol (Figure RTR2). In addition, we found that the RNA is degraded by RNases in the sera. In the figure below, we show an iterative imaging with *GRSF1* and FISH, which worked the best.

Figure RTR2: Iterative imaging of SABER-FISH with IF for *GRSF1*. Cells were treated for 5 hours with either EtBr or water. FISH was performed for *MT-CO1* (magenta) and *MT-ND6*

(cyan). The positions were saved and IF was performed for GRSF1 (yellow). Scale bars represent 5 μ m.

As can be seen, the signal is not comparable to our normal IF data (Fig. 3A, C), making it impossible to draw conclusions.

8. The authors should explain the rationale for using NDUFA7 as the sole control. Including multiple controls would strengthen the generality of the observed effects.

Response: Agreed. We now include c-MYC as another control.

9. For the TEM analysis presented in Figure EV3B, I recommend providing quantification to support claims of structural differences. A representative micrograph alone is generally insufficient, as mitochondrial ultrastructure can vary considerably.

Response: We thank the reviewer for the suggestion. We quantified the phenotypes as suggested and added a panel Fig. S6D.

10. The authors should investigate whether the inhibition granules contain only mRNAs or if they also include other RNA species.

Response: This is an interesting point. We have now added additional data for the antisense RNA of *MT-CO1*, *MT-ncCO1* (Figs. 2D,E. S5A,C,E and S3B,C). Thus, at least one noncoding RNA species is present in the inhibition granules.

Minor points:

1. The authors should clarify the rationale behind the differing formaldehyde fixation times used in various experiments.

Response: For our initial FISH experiments, we used published fixation times that were also suitable for immunofluorescence. However, we later optimized the fixation conditions and settled on 60 minutes.

2. NDUFA7 is misspelled in line 218.

Response: We have made the correction.

Reviewer #2 (Comments to the Authors (Required)):

Hansen et al. investigate the spatial distribution of mitochondrial RNAs (mt-RNAs) during stress conditions using single-molecule SABER-FISH. The authors report that transcription inhibition leads to the formation of a novel form of RNA granules in mitochondria, termed "inhibition granules." They further demonstrated that cells can recover from transcription inhibition by resuming transcription and redistributing mtRNA throughout the mitochondrial network. The authors conclude that inhibition granules may function as RNA reservoirs, potentially aiding in the restoration of mitochondrial function when stress is alleviated.

The topic of mitochondrial gene expression is interesting and important, as it directly influences cellular energy production and is linked to various diseases. The manuscript is generally interesting, the experiments are well-performed, and the text is well-written. We noticed that certain controls are missing and some conclusions are not fully supported by experimental data and would propose the authors to address a few points:

Response: We thank the referee for the comments and overall positive evaluation.

Major Comments:

- The authors claim that inhibition granules have a protective function mostly based on the bimodal kinetics of mtRNA degradation following transcription inhibition. While this is interesting and possible, at this point it is only correlative and not supported by experiments. The authors should experimentally test if preventing inhibition granules formation affects RNA stability, or, if the tools are not available, tempered their title and text to reflect that this is a proposed mechanism only.

Response: We agree. We changed the title and the text to temper our conclusions.

- Two control experiments would strengthen the findings from the authors on the existence of inhibition granules:

1. As an approach orthogonal to FISH, could the authors label mitochondrial transcripts to homogeneity with bromouridine (BrU), block transcription with EtBr, and look at inhibition granules after e.g. 5h with anti-BrU antibodies? This would also allow co-staining with antibodies against other intra-mitochondrial structures.

Response: Indeed, we tried to use EU-labeling with Click-Chemistry. However, the intensities for mitochondrial-encoded RNAs were low, and the data were hard to interpret, so we did not include these data in the manuscript (see Figure RTR1).

2. The authors should test for cross-reactivity of their probe with the mtDNA nucleoids specifically in EtBr-treated cells, where the general background of free mtRNA is dissipated, and strands of the mtDNA molecule might be exposed. Could the authors repeat their SABER-FISH + DNase-treatment, like in Fig. EV1A, but in EtBr-treated cells? It will be important to use a DNase that digests both ssDNA and dsDNA.

Response: To test for cross-reactivity, we performed iterative imaging with an ss/dsDNA antibody and *MT-ND6* FISH after EtBr treatment. An overlay of the FISH and IF signals show some overlap, as expected, due to the production of RNA on mtDNA and the limited resolution of light microscopy (Figure RTR3). However, the signal distribution does not show a 100% overlap, which would be the case if there were cross-reactivity.

Figure RTR3: Iterative imaging of SABER-FISH against *MT-ND6* and IF for ss/dsDNA. (A) Shown are maximum z-projections of the RNA-FISH, the IF and the merged data. (B) Line plot accessing the overlap. On the y-axis are shown the min-max normalized intensities for the FISH data (cyan) and the IF data (magenta). The distance is shown in μm . Vertical lines highlight intensity maxima.

- The effects of the treatments on RNA reorganization are inconsistent across the figures, and a better detection and quantification of the granules across some of the most important experiments would be helpful.

Response: We understand the concern. We have now added an additional quantification of granule formation throughout. As an additional unbiased way to quantify granule formation, we have measured the skewness of RNA signal distribution in cells (Fig. 2E, I, Fig. S5C, D). High skewness indicates the presence of granules, while low skewness indicates a more diffuse signal.

Minor Comments:

- Line 76: Dhir et al. (2018) do not mention the storage of dsRNA in granules.

Response: We thank the referee for pointing this out. The dsRNA granules were our own interpretation and we changed this now in the text stating: “Double stranded RNA (dsRNA) in mitochondria is also found in distinct foci (Dhir et al., 2018).”

- Depletion of SUV3, which induces the formation of dsRNA that can be stained by SABER-FISH, is not an ideal model for confirming that granule formation is triggered by transcription arrest. Downregulating POLRMT would be a better approach.

Response: We thank the reviewer for the comment. We already had perturbed POLRMT via IMT1B, and we have now added new data after 2'-CMA treatment, which also inhibits POLRMT (Fig. S5A).

- At the level of individual mRNAs, is there a correlation between the degradation kinetics (Figure 5) and the ability of mtRNA to form clusters?

Response: We compared our calculated early half-lives with the change in cellular area occupied and found no correlation (Fig. 5D).

Reviewer #3 (Comments to the Authors (Required)):

The authors have produced a neat manuscript to report that on various forms of mtDNA transcription inhibition, mt-mRNA species appear to be sequestered into aggregates that are distinct from the well described mitochondrial RNA granules. The data is well presented and the paper is very interesting. All expts are performed well and data analysis is good. The authors find that for most mt-mRNA species there are essentially two differing stabilities after transcriptional inhibition. I have several questions that do not detract from the high quality of this paper but the authors may wish to address them.

Response: We thank the referee for the positive evaluation of our manuscript.

First, do their data really show that subsets of the mt-mRNAs are sequestered into novel granules on transcriptional inhibition? Is it not possible that all transcripts pass through this intermediate granule that is a natural step in expression? It is merely that inhibition leads to the transcripts remaining in this intermediate stage, possibly as a consequence of the stress?

Response: This is an interesting question. We cannot exclude the possibility that the inhibition granules are stabilized intermediate granules. We now discuss this possibility.

Could it be simply that the transcript remains associated with the stalled or incomplete mitoribosome? It would be very interesting to know whether this stress intermediate can be labelled with anti-mitoribosome antibodies.

Response: It seems unlikely that the observed granules are due to stalled ribosomes that might cluster, as one would expect at least a similar phenotype with the mitochondrial translation inhibitor, Chloramphenicol, which we do not observe.

Or could the mt-mRNAs remain associated with SLRP/LRPPR prior to being transported to the mitoribosome? I guess what we're missing here is to have some idea/marker as to what this intermediate may be? Are the stored transcripts translated? Irrespective, its a very neat piece of work.

Response: We do not observe the clustering of LRPPRC by IF after transcription inhibition, so if the granule represents an intermediate state, it is unlikely to involve LRPPRC. In future work, it will be important to investigate whether inhibition granules represent an intermediate state where RNAs are stored before transitioning to mitochondrial ribosomes for translation.

May 28, 2025

RE: Life Science Alliance Manuscript #LSA-2024-03082-TR

Prof. L. Stirling Churchman
Harvard Medical School
Department of Genetics
77 avenue louis pasteur
New Research Building, 356
Boston, MA 2139

Dear Dr. Churchman,

Thank you for submitting your revised manuscript entitled "Transcription arrest induces formation of RNA granules in mitochondria". As you will see below, reviewers are satisfied with the revisions in place. Reviewer 2 suggested further tempering some language. While the summary blurb and abstract do not need revision, we concur that the section title on line 221 should be aligned with the claims made in this section ("mt-mRNA degradation kinetics suggest protection by RNA granules" or similar). We would be happy to publish your paper in Life Science Alliance pending this change and final revisions necessary to meet our formatting guidelines.

- Please add the X and Bluesky handles of your host institute/organization, as well as your own and/or one of the authors, to our system.
- Please upload a clean manuscript file, without the colored text.
- Please add a Conflict of Interest statement to your main manuscript text.
- Please add a callout for Figure 2I to your main manuscript text.

A. FINAL FILES:

B. MANUSCRIPT ORGANIZATION AND FORMATTING:

spreadsheets for the main figures of the manuscript. If you would like to add source data, we would welcome one PDF/Excel-file per figure for this information. These files will be linked as supplementary "Source Data" files.

Sincerely,

Reviewer #1 (Comments to the Authors (Required)):

The authors have made an effort to respond to most previous criticisms. However, several key concepts could not be demonstrated (e.g., the differentiation between RNA granules and inhibition granules, or the availability of mRNAs for translation after stress), which reduces the impact of the story from what it originally promised.

Reviewer #2 (Comments to the Authors (Required)):

The authors have addressed most of my concerns. I appreciate that the authors changed their title to reflect the descriptive nature of their observation.

Similarly, I would strongly encourage the authors to temper their claims in the summary Blurb (line 18 "that show protectivity for transcripts"), the abstract (line 30/31 "Inhibition granules appear to stabilize...") and in the manuscript (title line 221 "Inhibition granules stabilize mt-mRNAs"), since those statements suggest causality, which is not currently supported by the data.

The authors may also want to cite earlier work reporting the existence of stress granules in chloroplasts PMID: 18710928.

Reviewer #3 (Comments to the Authors (Required)):

I thank the authors for their response to my comments and am satisfied.

June 2, 2025

RE: Life Science Alliance Manuscript #LSA-2024-03082-TRR

Prof. L. Stirling Churchman
Harvard Medical School
Department of Genetics
77 avenue louis pasteur
New Research Building, 356
Boston, MA 2139

Dear Dr. Churchman,

Thank you for submitting your Research Article entitled "Transcription arrest induces formation of RNA granules in mitochondria". It is a pleasure to let you know that your manuscript is now accepted for publication in Life Science Alliance. Congratulations on this interesting work.

DISTRIBUTION OF MATERIALS:

Again, congratulations on a very nice paper. I hope you found the review process to be constructive and are pleased with how the manuscript was handled editorially. We look forward to future exciting submissions from your lab.

Sincerely,
